# INFLUENCE-BASED ATTRIBUTIONS CAN BE MANIPULATED

## ABSTRACT

Influence Functions are a standard tool for attributing predictions to training data in a principled manner and are widely used in applications such as data valuation and fairness. In this work, we present realistic incentives to manipulate influence-based attributions and investigate whether these attributions can be *systematically* tampered by an adversary. We show that this is indeed possible for logistic regression models trained on ResNet feature embeddings and standard tabular fairness datasets and provide efficient attacks with backward-friendly implementations. Our work raises questions on the reliability of influence-based attributions in adversarial circumstances.

## 1 INTRODUCTION

Influence Functions are a popular tool for data attribution and have been widely used in many applications such as data valuation (Richardson et al., 2019; Hesse et al., 2023; Sundararajan & Krichene, 2023; Jia et al., 2019), data filtering/subsampling/cleaning (Wu et al., 2022; Wang et al., 2020; Miao et al., 2021; Teso et al., 2021; Meng et al., 2022), fairness (Li & Liu, 2022; Wang et al., 2024; Sattigeri et al., 2022; Kong et al., 2021; Pang et al., 2024; Chhabra et al., 2023; Chen et al., 2024; Yao & Liu, 2023; Ghosh et al., 2023) and so on. While earlier they were being used for benign debugging, many of these newer applications involve adversarial scenarios where participants have an incentive to manipulate influence scores; for example, in data valuation a higher monetary sum is given to samples with a higher influence score and since good data is hard to collect, there is an incentive to superficially raise influence scores for existing data. Thus, an understanding of whether and how influence functions can be manipulated is essential to determine their proper usage and for putting guardrails in place. While a lot of work in the literature has studied manipulation of feature-based attributions (Heo et al., 2019; Anders et al., 2020; Slack et al., 2020), whether data attribution methods, specifically influence functions, can be manipulated has not been explored. To this end, our paper investigates the question and shows that it is indeed possible to *systematically* manipulate influence-based attributions according to the manipulator's incentives.

**Simply put, we show that it is possible to systematically train a malicious model very similar to the honest model in test accuracy but has desired influence scores**. To formalize the setup we divide the function pipeline in terms of two entities – Data Provider who provides training data and Influence Calculator who trains a model on this data and finds the influence of each training sample on model predictions. Out of these, Influence Calculator is considered to be the adversary who wishes to change the influence scores for some training samples and does so covertly by training a malicious model which is indistinguishable from the original model in terms of test accuracy but leads to desired influence scores. This setting captures two important downstream applications where incentives are meaningful: data valuation, where the adversary has an incentive to raise influence scores for monetary gain and fairness, where the adversary wants to manipulate influence scores for reducing the fairness of a downstream model.

We next define and provide algorithms to carry out two kinds of attacks in this setup: Targeted and Untargeted. Targeted attacks are for the data valuation application and specifically manipulate influence scores for certain target samples. The primary challenge with these attacks is that calculating gradients of influence-based loss objectives is highly computationally infeasible. We address this challenge by proposing a memory-time efficient and backward-friendly algorithm to compute the gradients while using existing PyTorch machinery for implementation. This contribution is of

independent technical interest, as the literature has only focused on making forward computation of influence functions feasible, while we study techniques to make the *backward pass* viable. Our algorithm brings down the memory required for one forward + backward pass from not being feasible to run on a 12GB GPU to 7GB for a 206K parameter model and from 8GB to 1.7GB for a 5K model.

Experiments on multiclass logistic regression models trained on ResNet50 features show that our targeted attacks achieve a high success rate, a maximum of 94%, without much accuracy drop across three datasets. One final question that comes to mind is – is it always possible to manipulate the influence scores for any given training sample? Using a theoretical construction, we give an impossibility theorem which states that there exist samples for which the influence score cannot be manipulated irrespective of the model, making this a property of the data rather than the model.

Untargeted attacks are for the fairness application and unlike targeted attacks, manipulate influence scores arbitrarily without targeting specific samples. We find that surprisingly enough scaling model weights is a good enough strategy for such attacks without changing model accuracy. In our experiments on standard tabular fairness datasets, we observe that due to influence score manipulation fairness of downstream models is affected a lot, leading to a maximum of 16% difference in fairness metric with and without influence manipulation.

Summarizing, we formalize a setup for systematically manipulating influence-based attributions and instantiate it for data valuation and fairness use-cases, where adversarial incentives are involved. We provide algorithms for targeted and untargeted attacks on influence scores, and illustrate their efficacy experimentally. Our work exposes the susceptibility of influence-based attributions to manipulation and highlights the need for careful consideration when using them in adversarial contexts. This is akin to what has been previously observed for feature attributions Bordt et al. (2022).

## 2   PRELIMINARIES

Consider a classification task with an input space $\mathcal{X} = \mathbb{R}^d$ and labels in set $\mathcal{Y}$. Let the training set of size $n$ be denoted by $Z_{\text{train}} = \{z_i\}_{i=1}^n$ where each sample $z_i$ is an input-label pair, $z_i = (x_i, y_i) \in \mathcal{X} \times \mathcal{Y}$. Let the loss function at a particular sample $z$ and model parameters $\theta \in \Theta$ be denoted by $L(z, \theta)$. Using the loss function and the training set, a model parameterized by $\theta \in \Theta$ is learnt through empirical risk minimization, resulting in the optimal parameters $\theta^\star := \arg\min_{\theta \in \Theta} \frac{1}{n} \sum_{i=1}^n L(z_i, \theta)$. The gradient of the loss w.r.t. parameters $\theta$ for the minimizer at a sample $z$ is given by $\nabla_\theta L(z, \theta^\star)$. Hessian of the loss for the minimizer is denoted by $H_{\theta^\star} := \frac{1}{n} \sum_{i=1}^n \nabla_\theta^2 L(z_i, \theta^\star)$. For brevity, we call the model parameterized by $\theta$ as model $\theta$. Next we give the definition of Influence Functions used in our paper.

**Definition 1** (Influence Function Koh & Liang (2017)). *Assuming that the empirical risk is twice-differentiable and strictly convex in model parameters $\theta$, the influence of a training point $z$ on the loss at a test point $z_{\text{test}}$ is given by,*

$$\mathcal{I}_{\theta^\star}(z, z_{\text{test}}) := -\nabla_\theta L(z_{\text{test}}, \theta^\star)^\top H_{\theta^\star}^{-1} \nabla_\theta L(z, \theta^\star) \tag{1}$$

*where $\nabla_\theta L(z_{\text{test}}, \theta^\star)$ and $\nabla_\theta L(z, \theta^\star)$ denote the loss gradients at $z_{\text{test}}$ and $z$ respectively, while $H_{\theta^\star}^{-1}$ denotes the hessian inverse.*

For logistic regression, the influence function has a closed form given by,

$$\mathcal{I}_\theta(z, z_{\text{test}}) = -y_{\text{test}} y \cdot \sigma(-y_{\text{test}} \theta^\top x_{\text{test}}) \cdot \sigma(-y \theta^\top x) \cdot x_{\text{test}}^\top H_\theta^{-1} x \tag{2}$$

where $y \in \{-1, 1\}$ and $\sigma(t) = \frac{1}{1+\exp(-t)}$ Koh & Liang (2017).

Given a test set of size $m$, $Z_{\text{test}} = \{z_{\text{test\_i}}\}_{i=1}^m$, we define the overall influence of a training point $z$ on the loss of the test *set* to be the sum of its influence on all test points $z_{\text{test\_i}}$ individually, written as

$$\mathcal{I}_{\theta^\star}(z, Z_{\text{test}}) := \sum_{i=1}^m \mathcal{I}_{\theta^\star}(z, z_{\text{test\_i}}) \tag{3}$$

The difference between Eq.1 and Eq. 3, that is whether the influence is calculated on a single test point vs. a test set, is understood from context.

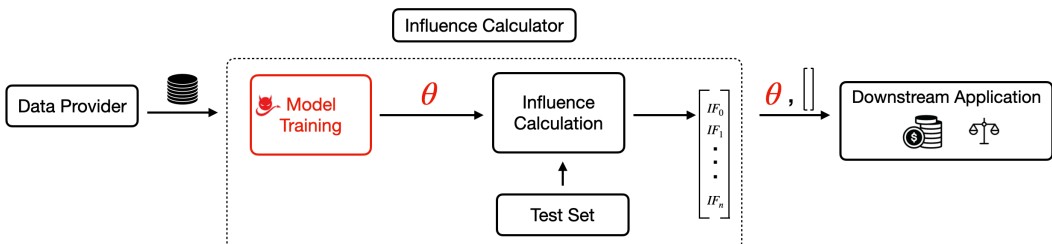

Figure 1: Threat Model. Data Provider provides training data. Influence Calculator trains a model and computes influence scores for the training data on the trained model and a test set. It outputs both the trained model and the resulting influence scores, which are used for a downstream application such as data valuation or fairness. Adversarial manipulation happens in the model training process, which trains a malicious model to achieve desired influence scores, while maintaining similar accuracy as the honest model.

## 3 GENERAL THREAT MODEL

In this section, we give a description of our setup and general threat model. We later instantiate these with two downstream applications, data valuation and fairness, where the objectives and incentives differ. *While we ground our discussion on these two applications, the attacks or their slight variations can apply to other applications.*

**Setup.** The standard influence function pipeline comprises of two entities: a Data Provider and an Influence Calculator. **Data Provider** holds all the training data privately and supplies it to the Influence Calculator. **Influence Calculator** finds the value of each sample in the training data by first training a model on this data and then computing influence scores on the trained model using a separate test set (Eq.3). We assume that the test set comes from the same underlying distribution as the training data. Influence Calculator outputs the trained model and the influence scores of each training sample ranked in a decreasing order of influence scores. These rankings/scores are then used for a **downstream application**.

An adversary who has incentives in the downstream application, would want to send manipulated influence scores to the downstream application. Now the question is, in which part of the IF pipeline should the adversarial manipulation occur? Turning to prior work on manipulating feature attributions (Anders et al., 2020; Slack et al., 2020; Heo et al., 2019; Pruthi et al., 2019), the popular choice has been to corrupt the model training process. In these attacks the compromised model training process outputs a malicious model which simultaneously has desired influence scores and is similar to the unaltered original model in test accuracy, thereby making the two models indistinguishable w.r.t. test predictions. Such an attack cannot be detected without access to the training pipeline or logs, making it the popular choice for manipulating explanations. Motivated by this, we attack the model training process in our paper and specify the resultant threat model next.

**General Threat Model.** We consider the training data held by the data provider and the test set used by the influence calculator to be fixed. We also assume the influence calculation process to be honest. The adversarial manipulation to maliciously change influence scores for some training samples happens during model training. To achieve this, the compromised model training process outputs a malicious model $\theta'$ such that $\theta'$ leads to desired influence scores but has similar test accuracy as original honest model $\theta^\star$.

**Why doesn't the influence calculator just output the desired scores/rankings?** A natural technique that comes to mind for manipulation of influences scores is to simply output the desired scores/rankings. This would be a viable attack only if the manipulation is discreet and cannot be detected; however an auditor with the ability to supply test samples can easily detect this manipulation (without access to training data) by checking the rank of the outputted influence matrix in only $O(d)$ queries where $d$ is the feature dimension. Kindly see Appendix Sec. A.0.1 for the detailed technique and proof. Intuitively speaking, since honest influence scores come from a closed form (even more so for logistic regression Eq. 2) and the fact that real-life learning tasks follow a structure, a lot of natural attacks in the influence calculation process might be detectable by an auditor with querying

abilities. An exploration and design of non-trivial working attacks in the influence calculation process makes for an interesting research direction and is left to future work.

> Takeaway : We will systematically train a malicious model which is very similar to the honest model in test accuracy, but has the desired influence scores/rankings.

## 4  DOWNSTREAM APPLICATION 1: DATA VALUATION

The goal of data valuation is to determine the contribution of each training sample to model training and accordingly assign a proportional monetary sum to each. One of the techniques to find this value is through influence functions, by ranking training samples according to their influence scores in a *decreasing order* (Richardson et al., 2019; Hesse et al., 2023; Sundararajan & Krichene, 2023; Jia et al., 2019). A higher influence ranking implies a more valuable sample, resulting in a higher monetary sum. Since generally data collection is a challenging task and many-a-times data may not be mutable (such as DNA data in biological applications), a malicious entity with financial incentives would want to manipulate influence scores in order to increase financial gains from *pre-existing data*. See App. Fig.6 for a pictorial representation of the data valuation setting.

**Threat Model.** The canonical setting of data valuation consists of 1) multiple data vendors and 2) influence calculator. Each vendor supplies a set of data; the collection of data from all vendors corresponds to the fixed training set of the data provider. The influence calculator is our adversary who can collude with data vendors while keeping the data fixed. The adversarial model training can change model parameters from $\theta^*$ to $\theta'$ while maintaining similar test accuracy as discussed in Sec.3.

**Goal of the adversary.** Given a set of target samples $Z_{\text{target}} \subset Z$, the goal of the adversary is to push the influence ranking of samples from $Z_{\text{target}}$ to top-$k$ or equivalently increase the influence score of samples from $Z_{\text{target}}$ beyond the remaining $n - k$ samples, where $k \in \mathbb{N}$.

**Single-Target Attack.** Let us first consider the case where $Z_{\text{target}}$ has only one element, $Z_{\text{target}} = \{z_{\text{target}}\}$. We formulate the adversary's attack as a constrained optimization problem where the objective function, $\ell_{\text{attack}}$, captures the intent to raise the influence ranking of the target sample to top-$k$ while the constraint function, dist, limits the distance between the original and manipulated model, so that the two models have similar test accuracies. The resulting optimization problem is given as follows, where $C \in \mathbb{R}$ is the model manipulation radius,

$$\min_{\theta':\text{dist}(\theta^\star,\theta')\leq C} \ell_{\text{attack}}(z_{\text{target}}, Z, Z_{\text{test}}, \theta') \tag{4}$$

**Multi-Target Attack.** When the target set consists of multiple target samples, $Z_{\text{target}} = \{z_{\text{target}_1}, z_{\text{target}_2} \cdots z_{\text{target}_q}\}$, the adversary's attack can be formulated as repeated applications of the Single-Target Attack, formally given as,

$$\min_{\theta':\text{dist}(\theta^\star,\theta')\leq C} \sum_{z_{\text{target}_i} \in Z_{\text{target}}} \ell_{\text{attack}}(z_{\text{target}_i}, Z, Z_{\text{test}}, \theta') \tag{5}$$

The actual objective used for both the attacks is given as, $\ell_{\text{attack}}(\cdot) = -\mathcal{I}_{\theta'}(z_{\text{target}}, Z_{\text{test}}) + \frac{1}{|S_{\theta'}|} \sum_{z \in S_{\theta'}} \mathcal{I}_{\theta'}(z, Z_{\text{test}})$ where $S_{\theta'} \subset Z_{\text{train}}$ contains all training samples $z$ s.t. $\mathcal{I}_{\theta'}(z, Z_{\text{test}}) > \mathcal{I}_{\theta'}(z_{\text{target}}, Z_{\text{test}})$ (see ablation study in Sec. 4.1) to understand why we chose this loss objective). Here the first term maximizes the influence of the target sample $z_{\text{target}}$ while the second term minimizes the influence of all samples which are currently more influential than $z_{\text{target}}$. Since this objective is non-convex, the optimization process results in local minima, which might be non-optimal. Therefore to get better results, we run every attack mutiple times, starting with random initializations of $\theta^\star$ in a radius $C$, as discussed later in the experiments (Sec.4.1).

**Efficient Backward Pass for Influence-based Objectives.** A natural algorithm to solve complicated optimization problems as our attacks in Eq. 4 & 5 is Gradient Descent, which involves a forward

and backward pass. However, for influence-based attack objectives, naive gradient descent is not feasible for either of the passes, mainly due to Hessian-Inverse-Vector Products (HIVPs) in the influence function definition which lead to a polynomial scaling of memory and time requirements w.r.t model parameters. Backward pass on our attack objectives is even harder as it involves *gradients* of influence-based loss objectives, making the attacks too expensive even for linear models trained on top of ResNet50 features used in our experiments where #parameters range from ~76k-206k.

While literature has studied ways to make the forward computation of influence functions efficient Schioppa et al. (2022); Guo et al. (2020); Koh & Liang (2017); Kwon et al. (2023), not much work has been done on making the *backward pass efficient*. To this end, we propose a simple technique – rewriting the original objective into a backward-friendly form – which renders the gradient computations efficient for influence-based objectives. This allows us to still use gradient descent and other existing machinery in PyTorch (Paszke et al., 2017). Our idea of rewriting the attack objective involves two essential steps : (1) linearizing the objective (2) making the linearized objective backward-friendly in PyTorch, as outlined in Alg.1. This algorithm is of independent technical interest and is generalizable to other use-cases where backward passes through HIVPs are needed. The complete algorithm for optimizing the loss with both forward and backward pass is elucidated in App. Alg. 3.

## 4.1 DATA VALUATION EXPERIMENTS

In this section, we investigate if the attacks we proposed for data valuation can succeed empirically. Specifically, we ask the following questions : (1) do our influence-based attacks perform better than a non-influence baseline?, (2) what is the behavior of our attacks w.r.t. different parameters such as radius $C$ and target set size? ,(3) what components contribute to the success of our attacks? and (4) lastly, can our attacks transfer to an unknown test set?. In what follows, we first explain our experimental setup and then discuss the results.

**Datasets & Models.** We use three standard image datasets for experimentation : CIFAR10 (Krizhevsky et al., 2009), Oxford-IIIT Pet (Parkhi et al., 2012) and Caltech-101 (Li et al., 2022). We split the respective test sets into two halves while maintaining the original class ratios for each. The first half is the test set shared between the model trainer and influence calculator used to optimize influence scores while the second is used as a pristine set for calculating the accuracy of models and also for transfer experiments discussed later. We pass all images through a pretrained ResNet50 model (He et al., 2016) from PyTorch to obtain feature vectors of size 2048 for and train linear models ($\theta^*$) on top of these features with cross-entropy loss and a learning rate of 0.001.

**Attack Setup & Evaluation.** The constraint function $\mathrm{dist}$ is set to L2-norm. Forward pass for the attacks is using the LiSSA Algorithm Koh & Liang (2017); details including parameters used can be found in App. Sec.A.1.2. For the Single-Target Attack, we randomly pick a training sample (which is not already in the top-$k$ influence rankings) as the target and carry out our attack on it. We repeat this process for 50 samples and report the fraction out of 50 which could be (individually) moved to top-$k$ in influence rankings as the success rate. To carry out the Multi-Target Attack, we randomly pick target sets of different sizes from the training set. The success rate now is the fraction of samples in the target set which could be moved top-$k$ in influence rankings. For many of our results, the success rates are reported under two regimes : (1) the high-accuracy similarity regime where the manipulated and original models are within 3% accuracy difference and (2) the best success rate irrespective of accuracy difference. We optimize every attack from 5 different initializations of $\theta^\star$ within a radius of $C$ and report the runs which eventually lead to the highest success rates.

**Baseline: Loss Reweighing Attack.** While our attacks are based on influence functions, we propose a non-influence baseline attack for increasing the importance of a training sample : reweigh the training loss, with a high weight on the loss for the target sample. We call this baseline the Loss Reweighing Attack, formally defined as, $\min_{\theta'} \sum_{z \in Z_{\mathrm{train}} \setminus \{z_{\mathrm{target}}\}} L(z; \theta') + \alpha \cdot L(z_{\mathrm{target}}; \theta')$, where $L$ is the model training loss and $\alpha \in \mathbb{R}$ is the weight on the loss of target sample. Intuitively, a larger weight $\alpha$ increases the influence of $z_{\mathrm{target}}$ on the final model, but results in a lower model accuracy and vice-versa. Directly reweighing the loss as in the baseline led to unstable training, so

Table 1: Success Rates of the Baseline vs. our Single-Target Attack for Data Valuation. $k$ is the ranking, as in top-$k$. $\Delta_{\mathrm{acc}} := \mathrm{TestAcc}(\theta^\star) - \mathrm{TestAcc}(\theta')$ represents drop in test accuracy for manipulated model $\theta'$. Two success rates are reported : (1) when $\Delta_{\mathrm{acc}} \leq 3\%$ (2) the best success rate irrespective of accuracy drop. (%) represents model accuracy. (-) means a model with non-zero success rate could not be found & hence accuracy can't be stated. *Our attack has a significantly higher success rate as compared to the baseline with a much smaller accuracy drop under all settings.*

| Dataset (Honest Model $\theta^\star$ Accuracy) | | CIFAR10 (89.8%) | | Oxford-IIIT Pet (92.2%) | | Caltech-101 (94.9%) | |
| --- | --- | --- | --- | --- | --- | --- | --- |
| Success Rate | | $\Delta_{\mathrm{acc}} \leq 3\%$ | Best ($\Delta_{\mathrm{acc}}$) | $\Delta_{\mathrm{acc}} \leq 3\%$ | Best ($\Delta_{\mathrm{acc}}$) | $\Delta_{\mathrm{acc}} \leq 3\%$ | Best ($\Delta_{\mathrm{acc}}$) |
| $k = 1$ | Baseline | 0.00 | 0.00 (-) | 0.00 | 0.00 (-) | 0.00 | 0.00 (-) |
| | Our | 0.64 | 0.90 (5.7%) | 0.88 | 0.94 (5.4%) | 0.74 | 0.85 (3.8%) |
| $k = 300$ | Baseline | 0.00 | 0.00 (-) | 0.10 | 1.00 (87.3%) | 0.08 | 0.84 (93.6%) |
| | Our | 0.76 | 0.90 (5.7%) | 0.88 | 0.94 (5.4%) | 0.74 | 0.85 (3.8%) |

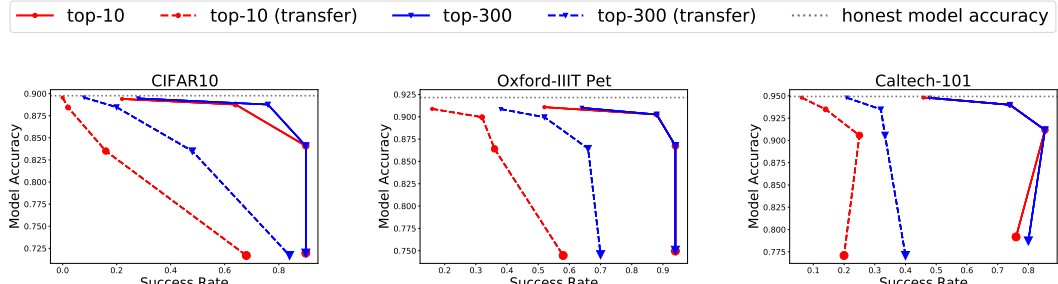

Figure 2: Behavior and Transfer results for Single-Target Attack in the Data Valuation use-case. Value of manipulation radius $C$ (Eq.4) increases from left to right in each curve. (1) Behavior on original test set (solid lines) : *As manipulation radius $C$ increases, manipulated model accuracy drops while attack success rate increases.* (2) Transfer on an unknown test set (dashed lines): *Success rate on an unknown test set gets better with increasing values of ranking $k$.*

we instead implemented the baseline with weighted sampling according to weight $\alpha$ in each batch (rather than uniform sampling).

For more experimental details, kindly refer to the Appendix Sec. A.1.2. Next we discuss our results.

**Our Single-Target attack performs better than the Baseline.** As demonstrated in Table 1, our influence-based attacks indeed performs better than the baseline – while the baseline has a low success rate across the board, our attack achieves a success rate of 64-88% in the high accuracy regime and 85-94% without accuracy constraints. The baseline is able to achieve a high success rate when ranking $k$ is large, but only with a massive accuracy drop. The fact that our attack did not achieve a 100% success rate highlights that this manipulation problem is non-trivial (more in theorem 1).

**Behavior of our Single-Target attack w.r.t manipulation radius $C$ & training set size.** Theoretically, the manipulation radius parameter $C$ in our attack objectives (Eq. 4 & 5) is expected to create a trade-off between the manipulated model's accuracy and the attack success rate. Increasing $C$ should result in a higher success rate as the manipulated model is allowed to diverge more from the (optimal) original model but on the other hand its accuracy should drop and vice-versa. We observe this trade-off for all three datasets and different values of ranking $k$, as shown in Fig.2 (solid lines).

We also anticipate our attack to work better with smaller training sets, as there will be fewer samples competing for top-$k$ rankings. Experimentally, this is found to be true – Pet dataset with the smallest training set has the highest success rates, as shown in Fig.2 & Table 1.

**Our attacks transfer when influence scores are computed with an unknown test set.** When an unknown test set is used to compute influence scores, our attacks perform better as ranking $k$ increases, as shown in Fig.2. This occurs because rank of the target sample, optimized with the

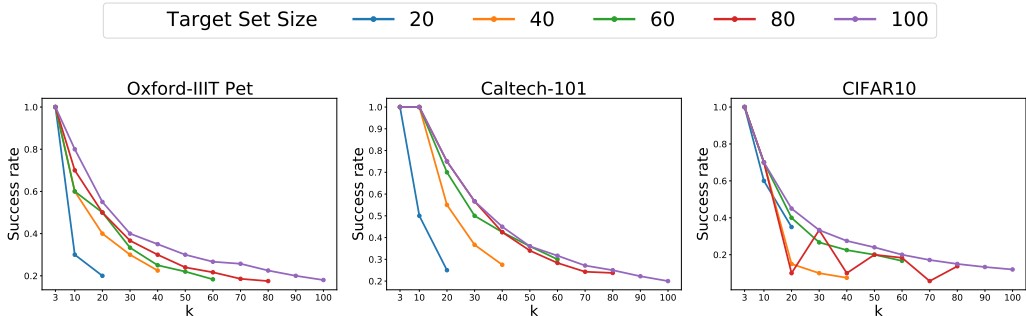

Figure 3: Performance of Multi-Target Attack in the Data Valuation use-case. Results for the high-accuracy regime. *Success Rates are higher when target set size is greater than the desired ranking k.*

original test set, deteriorates with the unknown test set and a larger $k$ increases the likelihood of the target still being in the top-$k$ rankings.

Next we discuss results for the Multi-Target Attack scenario, where the target is not a single training sample, but rather a collection of multiple training samples. We investigate the following question.

**How does our Multi-Target Attack perform with changing target set size and desired ranking $k$?** Intuitively, our attack should perform better when the size of the target set is larger compared to ranking $k$ – this is simply because a larger target set offers more candidates to take the top-$k$ rankings spots, thus increasing the chances of some of them making it to top-$k$. Our experimental results confirm this intuition; as demonstrated in Fig.3, we observe that (1) for a fixed value of ranking $k$, a larger target set size leads to a higher success rate; target set size of 100 has the highest success rates for all values of ranking $k$ across the board, and (2) the success rate decreases with increasing value of $k$ for all target set sizes and datasets. These results are for the high-accuracy similarity regime where the original and manipulated model accuracy differ by less than $3\%$.

**Easy vs. Hard Samples.** We find that target samples which rank very high or low in the original influence rankings are easier to push to top-$k$ rankings upon manipulation (or equivalently samples which have a high magnitude of influence either positive or negative). This is so because the influence scores of extreme rank samples are more sensitive to model parameters as shown experimentally in Fig. 4 and App. Fig.7, thus making them more susceptible to influence-based attacks.

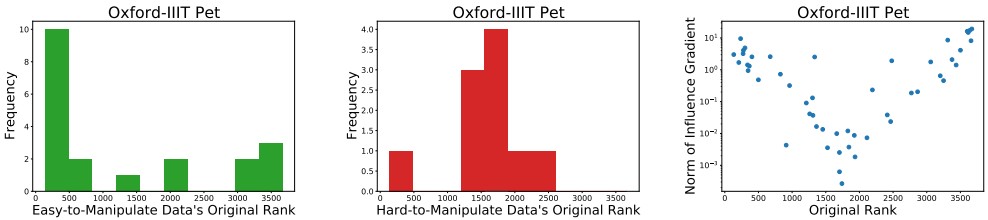

Figure 4: Histograms for original ranks of easy-to-manipulate samples (L), that of hard-to-manipulate samples (M), scatterplots for influence gradient norm vs. original ranks of (R) 50 random target samples. Ranking $k := 1$. For other datasets, see App. Fig.7.

**Imposibility Theorem for Data Valuation Attacks.** We observe in Fig.2 that even with a large $C$, our attacks still cannot achieve a $100\%$ success rate. Motivated by this, we wonder if there exist target samples for which the influence score cannot be moved to top-$k$ rank? The answer is yes and we formally state this impossibility result as follows, with the proof in Appendix Sec. A.1.3.

> **Theorem 1.** *For a logistic regression family of models and any target influence ranking $k \in \mathbb{N}$, there exists a training set $Z_{\text{train}}$, test set $Z_{\text{test}}$ and target sample $z_{\text{target}} \in Z_{\text{train}}$, such that no model in the family can have the target sample $z_{\text{target}}$ in top-$k$ influence rankings.*

**Ablation Study : What components contribute to the success of our attack?** Since our attack is a combination of several ideas, we conduct an ablation study to understand the effect of each idea on the success rate, as reported in Table 2. The different ideas are as follows.

- Maximize the target data's influence: Given a target sample, the simplest idea to move it to top-$k$ influence rankings is to maximize its own influence score, objective written as $\max_{\theta':\text{dist}(\theta^*,\theta')\leq C} \mathcal{I}_{\theta'}(z_{\text{target}}, Z_{\text{test}})$. This attack doesn't achieve high success rates, even without accuracy constraints which could be due to an inherent drawback : this objective doesn't consider other training samples' influence scores.

- \+ Minimize the influence of samples that are ranked top-$k$: Instead of just increasing the influence score of the target sample, this objective also lowers the score for the samples currently ranked top-$k$, given as $\max_{\theta':\text{dist}(\theta^*,\theta')\leq C} \mathcal{I}_{\theta'}(z_{\text{target}}, Z_{\text{test}}) - \frac{1}{K}\sum_{z:\text{rank of } z \leq k} \mathcal{I}_{\theta'}(z, Z_{\text{test}})$. We observe empirically that the optimization procedure of this objective gets stuck in local minima easily.

- \+ **(Our objective)** Minimize the influence score of all samples whose influence is larger than that of the target sample : This is the final objective used by us and lowers the influence of all training samples which have a higher influence than that of the target sample instead of just the top-$k$ (as in the previous objective), $\min_{\theta':\text{dist}(\theta^*,\theta')\leq C} \frac{1}{|S_{\theta'}|}\sum_{z \in S_{\theta'}} \mathcal{I}_{\theta'}(z, Z_{\text{test}}) - \mathcal{I}_{\theta'}(z_{\text{target}}, Z_{\text{test}})$ where $S_{\theta'} \subseteq Z_{\text{train}}$ has all training samples $z$ s.t. $\mathcal{I}_{\theta'}(z, Z_{\text{test}}) > \mathcal{I}_{\theta'}(z_{\text{target}}, Z_{\text{test}})$. Empirically, we find that this objective function decreases the chance of being stuck at suboptimal solutions and the loss keeps reducing throughout the optimization trajectory resulting in higher success rates.

- \+ **(Our final attack)** Multiple random initializations. Because the above objective function is non-convex, we find that using multiple random initializations of the honest model within a radius $C$ helps to obtain a better solution, especially with a larger value of parameter $C$, when the search space is bigger. As a result, we observe significant improvement in terms of 'best' success rates (where $C$ can be very large). This is our final attack.

Table 2: Ablation study for Single-Target Attack in Data Valuation. Ranking $k := 10$. $\Delta_{\text{acc}} := \text{TestAcc}(\theta^\star) - \text{TestAcc}(\theta')$ represents the drop in test accuracy for a manipulated model. (%) represents model accuracy. Two success rates are reported : (1) when $\Delta_{\text{acc}} \leq 3\%$ and (2) the best success rate irrespective of accuracy drop. *Our final objective with multiple random initializations of original model within radius $C$ leads to highest success rates.*

| Dataset (Honest Model $\theta^\star$ Accuracy) | CIFAR10 (89.8%) | | Oxford-IIIT Pet (92.2%) | | Caltech-101 (94.9%) | |
| Success Rate | $\Delta_{\text{acc}} \leq 3\%$ | Best ($\Delta_{\text{acc}}$) | $\Delta_{\text{acc}} \leq 3\%$ | Best ($\Delta_{\text{acc}}$) | $\Delta_{\text{acc}} \leq 3\%$ | Best ($\Delta_{\text{acc}}$) |
|---|---|---|---|---|---|---|
| Max. Target Inf. | 0.44 | 0.64 (15.9%) | 0.64 | 0.70 (15.2%) | 0.52 | 0.72 (11.7%) |
| + Min. Top-$k$ Inf. | 0.60 | 0.72 (20.0%) | 0.74 | 0.74 (1.6%) | 0.68 | 0.68 (3.6%) |
| + Min. Higher-Rank Inf. | 0.60 | 0.80 (6.0%) | 0.88 | 0.88 (2.1%) | 0.74 | 0.77 (8.3%) |
| + Multiple Rand Init. (**Ours**) | **0.64** | **0.90** (5.5%) | **0.88** | **0.94** (5.3%) | **0.74** | **0.85** (7.1%) |

## 5 DOWNSTREAM APPLICATION 2: FAIRNESS

Recently, a lot of studies have used influence functions in different ways to achieve fair models (Li & Liu, 2022; Wang et al., 2024; Sattigeri et al., 2022; Kong et al., 2021; Pang et al., 2024; Chhabra et al., 2023; Chen et al., 2024; Yao & Liu, 2023; Ghosh et al., 2023). In our paper, we focus on the study by Li & Liu (2022) as they use the same definition of influence functions as us. The suggested approach in Li & Liu (2022) to achieve a fair model is by *reweighing training data* based on influence scores for a *base model* and then using this reweighed data to train a new downstream model from

scratch. This downstream model is expected to have high fairness as a result of the reweighing. For a pictorial representation of this process, see App. Fig. 8. The weights for training data are found by solving an optimization problem which places influence functions in the constraints. See Appendix Sec.A.2 for more details on the optimization problem.

Ultimately, weights of the training data determine the fairness of the downstream model. Since these weights are derived from influence scores, manipulating the influence scores can alter the fairness of the downstream model. As a result, a malicious entity who wants to spread unfairness is incentivized to manipulate influence scores.

**Threat Model.** Similar to the general setup, training and test set are fixed, influence calculator is assumed to be the adversary. Model trained by the influence calculator is now the *base model used by the reweighing pipeline.* The adversarial model training can tamper the *base model* parameters from $\theta^*$ to $\theta'$ to manipulate the influence scores while maintaining similar test accuracy.

**Goal of the adversary.** Fairness of the final downstream model is measured with a fairness metric. The concrete goal of the adversary is to make the value of this fairness metric worse for the downstream model than what could have been achieved without adversarial manipulation.

**Attack.** Since the goal of the adversary in this case is not tied to specific target samples, we propose an untargeted attack for the adversary. Our attack is deceptively simple – scale the base model $\theta^\star$ by a constant $\lambda > 0$. The malicious base model output by the model trainer is now $\theta' = \lambda \cdot \theta^\star$, instead of $\theta^\star$. Note that for logistic regression the malicious and original base model are indistinguishable since scaling with a positive constant maintains the sign of the predictions, leading to the same accuracy.

## 5.1 FAIRNESS MANIPULATION EXPERIMENTS

We investigate experimentally if our scaling attack can be successful in reducing fairness of the final downstream model. We use three standard fairness benchmarks : Adult Credit Becker & Kohavi (1996), German Credit Hofmann (1994) and Compas Angwin et al. (2022). For details about these datasets, see Appendix Table 3. We train L2-regularized logistic regression models for these datasets which act as base models for the training data reweighing pipeline. To carry out our fairness manipulation attack, we scale the models weights with different scaling coefficients and feed these scaled models to the reweighing pipeline. The reweighed training data is then used to train a final downstream model, which is expected to be fair. Parameter values for L2 regularization and the reweighing are same as in Li & Liu (2022) and can be found in Appendix Tables 4 and 3.

**Evaluation.** We evaluate the fairness of the final downstream model with a popular fairness metric, called demographic parity (DP) (See Appendix Def. 2 for definition). A higher value of DP gap implies less fairness and vice-versa. We call the attack a success if the final downstream model post-attack is less fair (higher DP gap) than the final model achieved without attacking. Additionally, we expect the test accuracies of the downstream model with and without attack to be similar to keep the manipulation discreet. Note that this is different from our threat model's indistinguishability constraint, which applies to the base model, not to the downstream model.

**Results.** As can be seen from our results in Fig.5, the scaling attack works surprisingly well across all datasets – downstream models achieved after our attack are considerably less fair (higher DP gap) than the models without attack, achieving a maximum difference of 16% in the DP gap. Simultaneously, downstream models post-attack maintain similar test accuracies to downstream models without attack. Since the process to achieve the downstream model involves a lot of steps, including solving a non-convex optimization problem to find training data weights and then retraining a model, we sometimes do not see a smooth monotonic trend in fairness metric values w.r.t. scaling coefficients. However, this does not matter much from the attacker's perspective as all the attacker needs is *one* scaling coefficient which meets the attack success criteria.

## 6 DISCUSSION ON SUSCEPTIBILITY AND DEFENSE

The susceptibility of influence functions to our attacks can come from the fact that there can exist models that behave very similarly (Rashomon Effect Rudin et al. (2024)) but have different influential samples up to an extent. Equivalently, changing the influence for many samples does not affect the model accuracy much, as is shown by our experiments (though there exist some samples for which

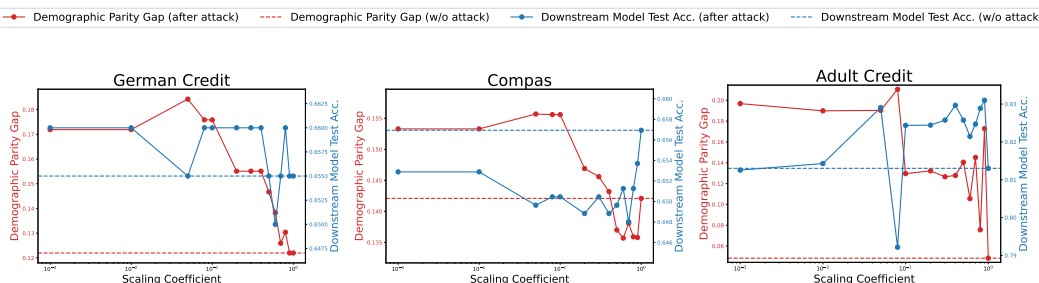

Figure 5: Scaling attack for the Fairness use-case. *Demographic Parity Gap of post-attack downstream models is higher than that of those w/o attack while test accuracies are comparable. This implies that post-attack downstream models are less fair than those w/o attacks.* Scaling coefficients in log scale.

the influence can't be manipulated, from theorem 1). Some plausible ways to defend against the attacks are (1) providing cryptographic proofs of honest model training using Zero-Knowledge Proofs Sun et al. (2023); Abbaszadeh et al. (2024) and, (2) to check if the model is atleast a local minima or not, since IFs assume that the model is an optimal solution to the optimization.

## 7    RELATED WORK

**Fragility of Influence Functions.** Influence functions proposed in Koh & Liang (2017) are an approximation to the effect of upweighting a training sample on the loss at a test point. This approximation error can be large as shown by (Basu et al., 2020; Bae et al., 2022; Epifano et al., 2023), making influence functions fragile especially for deep learning models. Our work is *orthogonal* to this line of work as we study the robustness of influence functions w.r.t. *model parameters* instead of approximation error of influence functions w.r.t. the true influence.

**Model Manipulation in the Threat Model.** Manipulating models to execute attacks is a prevalent theme in the literature. Slack et al. (2020) use model manipulations to corrupt feature attributions in tabular data while (Heo et al., 2019; Anders et al., 2020) do so in vision models. Pruthi et al. (2019) corrupt attention-based explanations for language models while maintaining model accuracy. Shahin Shamsabadi et al. (2022) show that it is possible to corrupt a fairness metric by manipulating an interpretable surrogate of a black-box model while maintaining empirical performance of the surrogate. Similar to these, our threat model also allows the adversary to manipulate models while maintaining the test accuracy. However, our adversarial goal is to corrupt influence-based attributions.

**Data Manipulation Attack on Explanations.** (Ghorbani et al., 2019; Alvarez Melis & Jaakkola, 2018; Zhang et al., 2020; Dombrowski et al., 2019; Kindermans et al., 2019) have studied how data can be manipulated to corrupt feature attributions. On the contrary, firstly, we keep data fixed and manipulate the model and secondly, we work with data attributions rather than feature attributions.

## 8    CONCLUSION & FUTURE WORK

While past work has mostly focused on feature attributions, in this paper we exhibit realistic incentives to manipulate data attributions. Motivated by the incentives, we propose attacks to manipulate outputs from a popular data attribution tool – Influence Functions. We demonstrate the success of our attacks experimentally on multiclass logistic regression models on ResNet features and standard tabular fairness datasets. Our work lays bare the vulnerablility of influence-based attributions to manipulation and serves as a cautionary tale when using them in adversarial circumstances.

While logistic regression is a good starting point for formulating and solving a new problem and these models are still relevant in many domains, we do think attacking influence functions for large models is an interesting avenue for future research. Some other future directions include exploring different threat models, additional use-cases and manipulating other kinds of data attribution tools.

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

# A APPENDIX

## A.0.1 AUDITING THE INFLUENCE CALCULATOR BY SUPPLYING TEST DATA

We provide an auditing algorithm that can detect if the influence calculator outputs arbitrary numbers as influence scores.

We first collect a sequence of test points $z_{\text{test\_1}}, \cdots, z_{\text{test\_d}}$ such that the rank of $G_{\text{test}}$ is $d$, where the $i$th row in $G_{\text{test}} \in \mathbb{R}^{d \times d}$ is $\nabla_\theta L(z_{\text{test}}, \theta^*)$ and $d$ is the size of model parameters $\theta^*$. This can be done because $\theta^*$ is publicly known by the auditor. We then query the influence calculator by feeding $z_{\text{test\_i}}$ one by one and collect the returned influence scores as $I \in \mathbb{R}^{m \times n}$. We compute the matrix $C := G_{\text{test}}^{-1} I$. Then we feed a new sequence of test points $z_{\text{test\_d+1}}, \cdots, z_{\text{test\_2d}}$ and suppose $I_i \in \mathbb{R}^m$ ($i = d + 1, \cdots, 2d$) are the returned influence scores. If there is any $i$ s.t. $I_i \neq \nabla_\theta L(z_{\text{test}}, \theta^*)^\top C$, we return *True*, i.e. state this influence calculator is malicious; otherwise, we return *False*.

We prove that if the influence calculator is honest, this algorithm will return False. In this case, $I = -G_{\text{test}} H_{\theta^*} G_{\text{train}}^\top$. Then the returned score $I_i = -\nabla_\theta L(z_{\text{test}}, \theta^*)^\top H_{\theta^*} G_{\text{train}}^\top = \nabla_\theta L(z_{\text{test}}, \theta^*)^\top G_{\text{test}}^{-1} \left(-G_{\text{test}} H_{\theta^*} G_{\text{train}}^\top\right) = -\nabla_\theta L(z_{\text{test}}, \theta^*)^\top G_{\text{test}}^{-1} I = \nabla_\theta L(z_{\text{test}}, \theta^*)^\top C$ should pass the auditing. In contrast, if the malicious influence calculator returns arbitrary scores, it will be captured by this auditing algorithm.

## A.1 DATA MANIPULATION ATTACK DETAILS

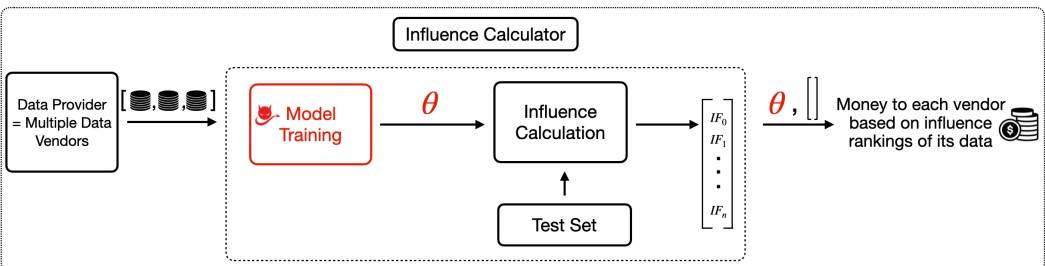

Figure 6: Data Valuation Setup and Threat Model.

## A.1.1 EFFICIENT BACKWARD PASS ALGORITHM

Our idea of rewriting the attack objective involves two essential steps : (1) linearizing the objective (2) making the linearized objective backward-friendly in PyTorch.

*Linearize the attack objective*: Generally the attack objective can be a non-linear combination of influence functions over different training samples, which makes the backward pass inefficient. Therefore we first transform the given objective into a linear combination of influence functions, $\hat{\ell}_{\text{attack}}((\mathcal{I}_\theta(z, Z_{\text{test}}) : z \in Z)) := u^\top (\mathcal{I}_\theta(z, Z_{\text{test}}) : z \in Z)$ for some vector $u \in \mathbb{R}^n$, such that, the objective and gradient values are the same, $\hat{\ell}_{\text{attack}}(\cdot) = \ell_{\text{attack}}(\cdot)$ and $\nabla_\theta \hat{\ell}_{\text{attack}}(\cdot) = \nabla_\theta \ell_{\text{attack}}(\cdot)$. Observe that from chain rule $\nabla_\theta \ell_{\text{attack}}(\mathcal{I}_\theta(z, Z_{\text{test}}) : z \in Z) = \sum_{z \in Z} \frac{\partial \ell_{\text{attack}}}{\partial \mathcal{I}_\theta(z, Z_{\text{test}})} \cdot \nabla_\theta \mathcal{I}_\theta(z, Z_{\text{test}})$.

Therefore, we can set $u$ as $\left(\frac{\partial \ell_{\text{attack}}}{\partial \mathcal{I}_\theta(z, Z_{\text{test}})} : z \in Z\right)$ while meeting the equal objective and gradient value requirement. Influence scores in this vector are computed using an efficient forward pass algorithm, given in Alg. 2.

*Get a PyTorch backward-friendly attack objective*: Simply expanding our linearized attack objective gives, $\hat{\ell}_{\text{attack}}(\cdot) = v_{\theta,1}^\top H_\theta^{-1} v_{\theta,2}$ where $v_{\theta,1} = \left(-\nabla_\theta \sum_{i=1}^m L\left(z_{\text{test\_i}}, \theta\right)\right)$ and $v_{\theta,2} = \left(\nabla_\theta \sum_{z \in Z} u_z \cdot L(z, \theta)\right)$. Gradient computations for this objective will have to go through HIVPs, which is highly inefficient. Therefore we next convert the linearized objective into one which does not involve HIVPs, again such that the objective and gradient values are same as the original objective.

Using chain rule, the gradient of the expanded linearized attack objective can be written as $\nabla_\theta \hat{\ell}_{\text{attack}}(\cdot) = (\nabla_\theta v_{\theta,1})^\top u_2 + u_1^\top (\nabla_\theta v_{\theta,2}) - u_1^\top (\nabla_\theta H_\theta) u_2$ where $u_1 = H_\theta^{-1} v_{\theta,1}$ and $u_2 = H_\theta^{-1} v_{\theta,2}$. PyTorch supports the gradient computation for functions of gradient, making $v_{\theta,1}$ and $v_{\theta,2}$ backward-friendly. PyTorch also calculates gradients for functions of hessian vector products *implicitly*, which leads to efficiency. Additionally, we can precompute $u_1$, $u_2$ and freeze them.

As a result, our final backward-friendly objective function is efficient and backward-friendly with PyTorch and is given as, $\bar{\ell}_{\text{attack}}(\theta) = v_{\theta,1}^\top u_2 + u_1^\top v_{\theta,2} - u_1^\top H_\theta u_2$. The algorithm for computing our backward-friendly objective $\bar{\ell}_{\text{attack}}$ is elucidated in Alg. 1.

**Derivation for expanding the linearized objective:**

$$
\begin{aligned}
\hat{\ell}_{\text{attack}}((\mathcal{I}_\theta(z, Z_{\text{test}}) : z \in Z)) &= u^\top (\mathcal{I}_\theta(z, Z_{\text{test}}) : z \in Z) \\
&= \sum_{z \in Z} u_z \cdot \mathcal{I}_\theta(z, Z_{\text{test}}) \\
&= \sum_{z \in Z} u_z \cdot \sum_{i=1}^m \mathcal{I}_\theta(z_{\text{test\_i}}, z) \\
&= \sum_{z \in Z} u_z \cdot \sum_{i=1}^m -\nabla_\theta L(z_{\text{test,i}}, \theta)^\top H_\theta^{-1} \nabla_\theta L(z, \theta) \\
&= \left( -\nabla_\theta \sum_{i=1}^m L(z_{\text{test,i}}, \theta) \right)^\top H_\theta^{-1} \left( \nabla_\theta \sum_{z \in Z} u_z \cdot L(z, \theta) \right) \\
&= v_{\theta,1}^\top H_\theta^{-1} v_{\theta,2}
\end{aligned}
$$

where $v_{\theta,1} = (-\nabla_\theta \sum_{i=1}^m L(z_{\text{test,i}}, \theta))$ and $v_{\theta,2} = (\nabla_\theta \sum_{z \in Z} u_z \cdot L(z, \theta))$.

**Chain Rule for gradient of expanded linearized objective:**

$$
\begin{aligned}
\nabla_\theta \hat{\ell}_{\text{attack}}((\mathcal{I}_\theta(z, Z_{\text{test}}) : z \in Z)) &= \nabla_\theta v_{\theta,1}^\top H_\theta^{-1} v_{\theta,2} \\
&= (\nabla_\theta v_{\theta,1})^\top \cdot H_\theta^{-1} v_{\theta,2} + v_{\theta,1}^\top H_\theta^{-1} (\nabla_\theta v_{\theta,2}) - v_{\theta,1}^\top H_\theta^{-1} (\nabla_\theta H_\theta) H_\theta^{-1} v_{\theta,2} \\
&= (\nabla_\theta v_{\theta,1})^\top u_2 + u_1^\top (\nabla_\theta v_{\theta,2}) - u_1^\top (\nabla_\theta H_\theta) u_2
\end{aligned}
$$

where $u_1 = H_\theta^{-1} v_{\theta,1}$ and $u_2 = H_\theta^{-1} v_{\theta,2}$.

---

**Algorithm 1** Get_Backward_Friendly_Attack_Objective

---

**Input:** Model Parameters $\theta$, Train Set $Z$, Test Set $Z_{\text{test}}$, Loss $L$, Original Attack Objective $\ell_{\text{attack}}$
**Output:** Backward-Friendly Attack Objective $\bar{\ell}_{\text{attack}}(\theta)$

1: Compute $(\mathcal{I}_\theta(z, Z_{\text{test}}) : z \in Z)$ from Appendix Alg. 2
2: Compute $u := \left( \frac{\partial \ell_{\text{attack}}}{\partial \mathcal{I}_\theta(z, Z_{\text{test}})} : z \in Z \right)$
3: Compute $v_{\theta,1} := (-\nabla_\theta \sum_{i=1}^m L(z_{\text{test\_i}}, \theta))$ and $v_{\theta,2} := (\nabla_\theta \sum_{z \in Z} u_z \cdot L(z, \theta))$
4: Compute and freeze $u_1 := H_\theta^{-1} v_{\theta,1}$ and $u_2 := H_\theta^{-1} v_{\theta,2}$
5: Compute $\bar{\ell}_{\text{attack}}(\theta) := v_{\theta,1}^\top u_1 + u_2^\top v_{\theta,2} - u_1^\top H_\theta u_2$
6: **Return:** $\bar{\ell}_{\text{attack}}(\theta)$

---

When using the original influence-based objective naively, it was not possible to even do one backward pass due to memory constraints. This algorithm brings down the memory required for one forward + backward pass from not being feasible to run on a 12GB GPU to 7GB for a 206K parameter model and from 8GB to 1.7GB for a 5K model.

---

**Algorithm 2** ForwardOnlyInf

**Input:** Parameters $\theta$, train set $Z$, test set $Z_{\text{test}}$, loss $L$
**Output:** $(\mathcal{I}_\theta(z, Z_{\text{test}}) : z \in Z)$

  1: Compute $L(Z_{\text{test}}, \theta) := \nabla_\theta \sum_{i=1}^{m} L(z_{\text{test\_i}}, \theta)$
  2: Compute $s_{\text{test}} := H_\theta^{-1} L(Z_{\text{test}}, \theta)$ by the hessian-inverse-vector product in Koh & Liang (2017).
  3: $\forall z \in Z$, compute $\mathcal{I}_\theta(z, Z_{\text{test}}) := s_{\text{test}}^\top \nabla_\theta L(z, \theta)$
  4: **Return:** $(\mathcal{I}_\theta(z, Z_{\text{test}}) : z \in Z)$

---

---

**Algorithm 3** Gradient-based optimization for Attack Loss $\ell_{\text{attack}}$

**Input:** Parameters $\theta^\star$, Radius $C$, train set $Z$, test set $Z_{\text{test}}$, loss $L$, attack objective $\ell_{\text{attack}}$, gradient-based optimizer $Opt$, distance function $\text{dist}$
**Output:** $\theta'$

  1: Set $\theta_0 :=$ Randomly chosen model parameters from a ball of radius $C$ centered around $\theta^\star$, $\mathcal{B}(\theta^\star, C)$ where distance is calculated using $\text{dist}$
  2: **for** $t = 1$ **to** $T$ **do**
  3:      $\bar{\ell}_{\text{attack}}(\theta_{t-1}) = \mathsf{Get\_Backward\_Friendly\_Attack\_Objective}(\theta_{t-1}, Z, Z_{\text{test}}, L, \ell_{\text{attack}})$
  4:      Compute $\nabla_\theta \bar{\ell}_{\text{attack}}(\theta_{t-1})$ through $\bar{\ell}_{\text{attack}}(\theta_{t-1}).\text{Backward}()$ in PyTorch
  5:      Update $\theta_{t-1} \rightarrow \theta_t$ by the given gradient-based optimizer $Opt$
  6:      If $\theta_t \notin \mathcal{B}(\theta^\star, C)$, clip $\theta_t$ to lie within $\mathcal{B}(\theta^\star, C)$ where distance is calculated using $\text{dist}$
  7: **end for**
  8: **Return:** $\theta' := \theta_T$

---

### A.1.2 EXPERIMENTAL DETAILS

**Dataset Details.** CIFAR10 (Krizhevsky et al., 2009) has a training/test set size of 50000/10000 with 10 output classes, Oxford-IIIT Pet (Parkhi et al., 2012) has a training/test set size of 3680/3669 with 37 output classes and Caltech-101 (Li et al., 2022) has a training/test set size of 6941/1736 with 101 output classes.

**Forward pass Details.** We use the LiSSA implementation given at `https://github.com/nimarb/pytorch_influence_functions` with recursion depth=1e6, scale=25 and damping factor= 0.01.

**Attack Details.** To optimize our attack objective, we use algorithm Alg. 3 for computing gradients with Adam as the optimizer Kingma & Ba (2014). We use two learning rates $\{0.01, 0.1\}$ and 100 steps of updates. We optimize every attack from 5 different initializations. We run our attacks with multiple values of the constraint radius $C = \{0.05, 0.1, 0.2, 0.5\}$. For each regime, the reported number is the highest we could obtain with different values of constants $C$ or $\alpha$.

**Baseline Details.** For training a model under the baseline attack we choose Adam as the optimizer, set the batch size as $256$ and update for $1400$ steps. We ran the baseline for different weights $\alpha$ with a logarithmic scaling from 10 to 1e18. We use weighted sampling instead of uniform to account for the weight on the target sample.

### A.1.3 PROOF OF IMPOSSIBILITY THEOREM (THEOREM 1)

*Proof of Theorem 1.* We first introduce a construction of 2-class classification dataset $Z_{\text{train}}, Z_{\text{test}} \subseteq \mathbb{R}^d \times \{1, -1\}$ and show under this construction and a logistic regression model $\theta$, there exists a target data $z_{\text{target}} \in Z_{\text{train}}$ such that no matter how we manipulate this linear model $\theta$, i.e. $\forall \theta \in \mathbb{R}^d$, the rank of $\mathcal{I}_\theta(z_{\text{target}}, Z_{\text{test}})$ among $(\mathcal{I}_\theta(z, Z_{\text{test}}) : z \in Z_{\text{train}})$ *cannot* reach top-1.

Denote $z_i \in \mathbb{R}^d$ as a one-hot vector which has all zeros except 1 at the $i$th dimension. We construct $Z_{\text{test}}$ as $\{(\mathbf{e}_1, 1)\}$, and construct $Z_{\text{train}} = \{(z_i, y_i) | i = 2, \cdots, d\} \cup \{(\mathbf{e}_1, 1), (-\mathbf{e}_1, 1)\}$ where $y_i$ can be arbitrarily selected from $\{1, -1\}$. By choosing $z_{\text{target}} = (\mathbf{e}_1, 1) \in Z_{\text{train}}$ and $z_{\text{bar}} = (-\mathbf{e}_1, 1) \in Z_{\text{train}}$, next we are going to prove $\mathcal{I}_\theta(z_{\text{target}}, Z_{\text{test}}) < \mathcal{I}_\theta(z_{\text{bar}}, Z_{\text{test}})$ for any $\theta \in \mathbb{R}^d$, which indicates $\mathcal{I}_\theta(z_{\text{target}}, Z_{\text{test}})$ among $(\mathcal{I}_\theta(z_{\text{target}}, Z_{\text{test}}) : z \in Z_{\text{train}})$ *cannot* reach top-1.

We adopt the computation of influence function on logistic regression from the original influence function paper (Koh & Liang, 2017). Denote $r(z_i, \theta) = \sigma(z_i^\top \theta) \cdot \sigma(-z_i^\top \theta)$ where $\sigma(t) = \frac{1}{1+\exp(-t)} > 0$. Then the Hessian $H_\theta$ of the logistic regression training loss $\frac{1}{d+1} \sum_{z \in Z_{\text{train}}} L(z, \theta)$ is $H_\theta = \frac{1}{d+1} \left( 2r(\mathbf{e}_1, \theta) \mathbf{e}_1 \mathbf{e}_1^\top + \sum_{i=2}^d r(z_i, \theta) z_i z_i^\top \right)$. Then we can calculate the influence function for $z_{\text{target}}$ and $z_{\text{bar}}$:

$$\mathcal{I}_\theta(z_{\text{target}}, Z_{\text{test}}) = -\sigma(-\theta^\top \mathbf{e}_1) \cdot \sigma(\theta^\top \mathbf{e}_1) \cdot \mathbf{e}_1^\top H_\theta^{-1} \mathbf{e}_1 = -\sigma(-\theta^\top \mathbf{e}_1) \cdot \sigma(\theta^\top \mathbf{e}_1) \cdot \frac{d+1}{r(\mathbf{e}_1, \theta)} < 0,$$

$$\mathcal{I}_\theta(z_{\text{bar}}, Z_{\text{test}}) = -\sigma(-\theta^\top \mathbf{e}_1) \cdot \sigma(\theta^\top \mathbf{e}_1) \mathbf{e}_1^\top H_\theta^{-1} (-\mathbf{e}_1) = \sigma(-\theta^\top \mathbf{e}_1) \cdot \sigma(\theta^\top \mathbf{e}_1) \cdot \frac{d+1}{r(\mathbf{e}_1, \theta)} > 0.$$

This complete the proof: $\forall \theta, \mathcal{I}_\theta(z_{\text{target}}, Z_{\text{test}}) < 0 < \mathcal{I}_\theta(z_{\text{bar}}, Z_{\text{test}})$ and therefore $z_{\text{target}}$ can never achieve top-1.

The above construction can be generalized to top-$K$ for any $K > 1$: we can construct $Z_{\text{test}}$ as $\{(\mathbf{e}_1, 1)\}$, and construct $Z_{\text{train}} = \{(z_i, y_i) | i = 2, \cdots, d\} \cup \{(\mathbf{e}_1, 1), K \text{ duplicated } (-\mathbf{e}_1, 1)\}$ where $y_i$ can be arbitrarily selected from $\{1, -1\}$. Then similar to the proof above, by choosing $z_{\text{target}} = (\mathbf{e}_1, 1) \in Z_{\text{train}}$ and $K$ duplicated training point in the training set $z_{\text{bar}}^k = (-\mathbf{e}_1, 1), k \in [K]$, we can prove $\forall \theta, \mathcal{I}_\theta(z_{\text{target}}, Z_{\text{test}}) < 0$ and $\forall \theta, k \in [K], \mathcal{I}_\theta(z_{\text{bar}}^k, Z_{\text{test}}) > 0$. Consequently, $\forall \theta, z_{\text{target}}$ will not achieve top-$K$.

$\square$

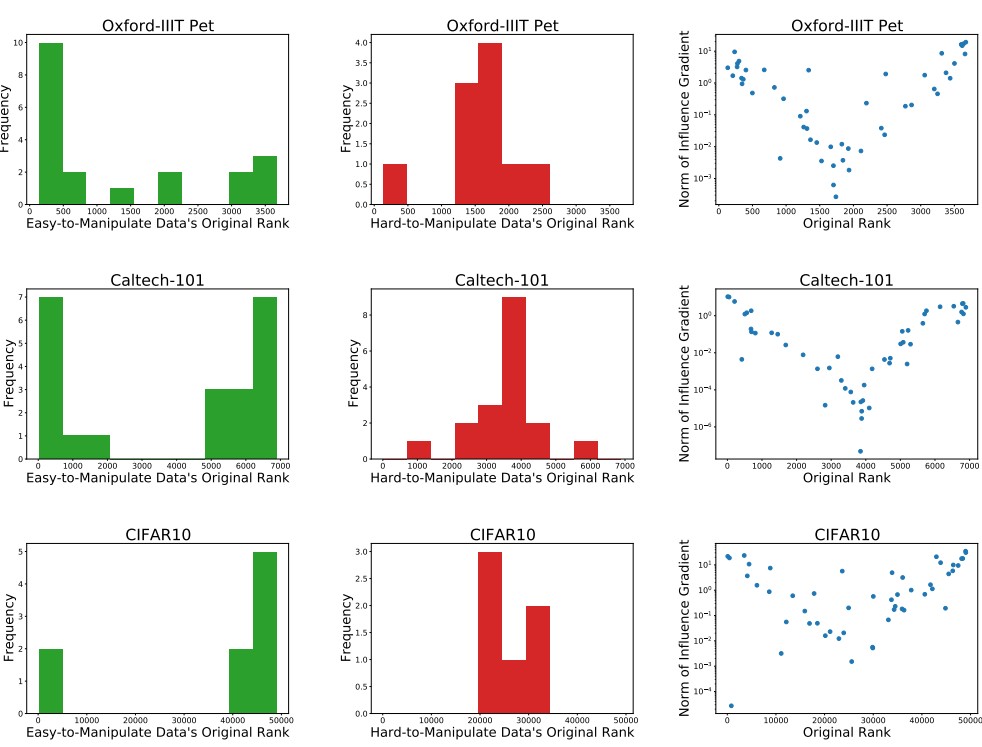

Figure 7: Histograms for original ranks of easy-to-manipulate samples, Histograms for original ranks of hard-to-manipulate samples, Scatterplots for influence gradient norm vs. original ranks of the 50 random target samples. Ranking $k := 1$. *Easy-to-manipulate samples have extreme original influence ranks (large positive or negative) as the samples with the extreme rankings also have higher influence gradient norms, where the gradient is taken w.r.t. model parameters.*

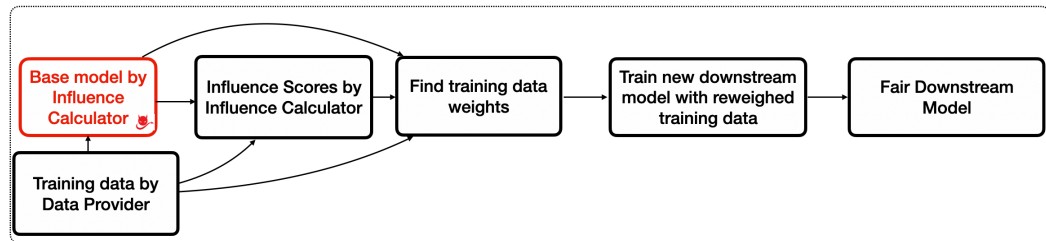

Figure 8: Overview of the process to achieve a fair model, as proposed by Li & Liu (2022). Our adversary, the model trainer manipulates the base model used to calculate influence scores.

## A.2 FAIRNESS MANIPULATION ATTACK DETAILS

**Optimization Problem for Reweighing Training Data as proposed by Li & Liu (2022)** is given as follows,

$$
\begin{aligned}
\text{minimize} \quad & \sum_i w_i \\
\text{subject to} \quad & \sum_i w_i \mathcal{I}_{\text{fair}}\,(z_i) = -f_{\text{fair}}^{\mathcal{V}} \\
& \sum_i w_i \mathcal{I}_{\text{util}}\,(z_i) \leq 0 \\
& w_i \in [0,1]
\end{aligned}
\tag{6}
$$

where $w_i$ refers to the weight of the $i^{th}$ training sample, $z_i$ refers to the $i^{th}$ training sample, $\mathcal{I}_{\text{util}}$ refers to our influence function, $\mathcal{I}_{\text{fair}}$ refers to some fairness influence function and $f_{\text{fair}}^{\mathcal{V}}$ corresponds to a differentiable fairness metric. In our threat model, the adversary manipulates the base model, which changes the influence scores $\mathcal{I}_{\text{util}}$.

An advanced version of the above optimization problem using additional parameters $(\beta, \gamma)$ which lead to various tradeoffs is given as,

$$
\begin{aligned}
\text{minimize} \quad & \sum_i w_i \\
\text{subject to} \quad & \sum_i w_i \mathcal{I}_{\text{fair}}\,(z_i) \leq -(1-\beta)\ell_{\text{fair}}^{\nu}\,, \\
& \sum_i w_i \mathcal{I}_{\text{util}}\,(z_i) \leq \gamma \left(\min_{\mathbf{v}} \sum_i v_i \mathcal{I}_{\text{util}}\,(z_i)\right), \\
& w_i \in [0,1].
\end{aligned}
\tag{7}
$$

**Fairness Metric.** We define the fairness metric used in our paper, Demographic Parity.

**Definition 2** (Demographic Parity Gap (DP) Dwork et al. (2012)). *Given a data distribution $\mathcal{D}$ over $\bar{\mathcal{X}} \times \{0,1\}$ from which features $x^{\backslash a}$ and sensitive attribute $x^a \in \{0,1\}$ are jointly drawn from, Demographic Parity gap for a model $f_\theta$ is defined to be the difference in the rate of positive predictions between the two groups, $|\Pr(\hat{y} \mid x^a = 0) - \Pr(\hat{y} \mid x^a = 1)|$ where $\hat{y}$ is the prediction $f_\theta(x^{\backslash a}, x^a)$.*

| Dataset | Predict | Train / Val. / Test Split | #Dim. | Sensitive Attribute | Group Pos. Rate |
|---|---|---|---|---|---|
| Adult | if annual income >= 50k | 22622/7540/15060 | 102 | Gender - Male / Female | 0.312/0.113 |
| Compas | if defendant rearrested in 2 yrs. | 3700/1234/1233 | 433 | Race - White / Non-white | 0.609/0.518 |
| German Credit | good/bad credit risk | 600/200/200 | 56 | Age - 30 | 0.742/0.643 |

Table 3: Details for datasets used in our Fairness Manipulation attack experiments. In our setting, the Validation set is the test set shared between the model trainer and influence calculator. The Test set is an untouched set which is used to assess the performance and fairness of the final model achieved after the training with the reweighed training set. All our results are reported on this untouched Test set. Table details borrowed from Li & Liu (2022).

| Dataset | Model $\ell_2$ reg. | DP $(\beta, \gamma)$ |
|---|---|---|
| Adult | 2.26 | (0.8,0.3) |
| Compas | 37.00 | (0.3,0.1) |
| German Credit | 5.85 | (0.5,0.0) |

Table 4: Details for the Fairness Manipulation attack experiments. We use the same values of parameters as used by Li & Liu (2022).

