# OpenReview forum: "Influence-based Attributions can be Manipulated"
_ICLR.cc/2025/Conference — Submitted to ICLR 2025_

### Official Review · Reviewer_jono · 2024-10-28

**Soundness:** 1
**Presentation:** 2
**Contribution:** 1
**Rating:** 3
**Confidence:** 3

**Summary:**

This paper proposed an attack to manipulate influence-based attributions by training a malicious model with slightly reduced accuracy, while achieving the attacked influence scores. Specifically, the authors aimed to find a malicious model $\theta’$ that maintained a similar accuracy to the original model $\theta$, which maximizing the influence score of the target sample. The authors also discussed the potential scenarios for the proposed attack, including data valuation and fairness use cases.

**Strengths:**

The authors investigated the vulnerability of influence-based attributions, which was an interesting direction.

**Weaknesses:**

1.	The main weakness was that the threat model on influence score was problematic. The influence score $I_\theta (z, z_{test})$ was computed based on a given model $\theta$ according to Eq. (1). When different model parameters $\theta$ were given, the influence score of a training point $z$ on the loss at a test point $z_{test}$ $I_\theta (z, z_{test})$ would indeed yield a different score. In other words, the discussion of the influence function was only meaningful within the context of the same model. Changes in model parameters would inevitably lead to variations in influence scores. Therefore, these variations should not be considered an attack on the influence function.

2.	The proposed attack scenarios were not reasonable. For instance, in data valuation tasks, why would an adversary be able to modify the model parameters? If the goal was to evaluate the importance of data for a specific model, the model parameters should remain fixed. For example, with large models like GPT-4, the adversary would not be able to alter the model parameters. Furthermore, if multiple models were used to evaluate the importance of data, why would the evaluator choose a malicious model over a set of trusted models? In other words, if the adversary cannot ensure that the evaluator uses the malicious model, then the proposed attack becomes impractical.

3.	The proposed attack constrained the L2 norm of the two parameters in Eq. (4) and in Lines 250-251 to ensure that the accuracies of the two models are similar. However, there is a lack of evidence that the L2 norm of parameters is related to the performance of the model. For example, consider a two-layer network with two categories $f(x) = Sigmoid(c\cdot Ax)$, where $c>0$. If $f(x)>0.5$, the class is 1; otherwise, the class is 0. In this case, changing the L2 norm of the parameters $B=c\cdot A$ (i.e., modifying $c$) does not affect the model's performance.

**Questions:**

1.	Why does $dist(\theta, \theta') <= C$ in Eq. (4) guarantee that the original and  malicious  models have similar accuracy? Given that different models with the same structure may have varying parameter magnitudes, how to determine the hyper-parameter $C$?

2.	Why does Eq. (3) compute the influence function of a training sample z on all test samples? Will the sum of the influence scores for all test samples cancel each other out, both positively and negatively?

---

> ### Author Response · Authors · 2024-11-20
>
> Dear Reviewer,
>
> Thanks for taking the time to review our paper, we deeply appreciate it.
>
> We are very glad that you find the idea of exposing vulnerabilities in influence functions interesting. Next we address your concerns.
>
> a) **“these variations should not be considered an attack on the influence function”** : What is an attack? An attack is a tool for an adversary to achieve its desired goal without getting caught. Attacks are meant to exploit some weakness or vulnerability otherwise how can a perfectly secure system be attacked?
> One of the vulnerabilities exploited by our proposed attack is the sensitivity of influence functions to model parameters – but this is no grounds to not call the method an attack! Infact our paper serves as a cautionary tale to not use sensitive methods (such as influence functions) for consequential tasks such as data valuation!
>
> b) Reasonable attack scenarios : Perhaps there are some misunderstandings, we clarify them one by one.
>
> *“why would an adversary be able to modify the model parameters?”*: Please note that the adversary in our case, influence calculator, is also responsible for training the model. For the attack, the adversary makes a few updates starting with the original honest model, which will be possible even for large models. Think of it like a company training some model and outputting both the influence scores (for compensating data providers) and a final model (which can be used by other entities such as other teams or customers).
>
> *“why would the evaluator choose a malicious model over a set of trusted models”*: In the absence of a clear idea of what is an ‘evaluator’ here is our best explanation : The setting where there exists a separate evaluator with access to multiple models is a totally different threat model and should be a different paper. While interesting, there could be multiple threat models and there is no one universal threat model which is applicable in all settings. Different circumstances call for different threat models.
> We work on one setting and propose one threat model. If you have any concerns on the threat model used in our paper, we would be happy to clarify them.
>
> c) **Contributions of our paper**: Note that ours is the first paper which introduces the problem of attacking attributions and proposes attacks with in-depth analysis. We believe the most important contribution of the paper is introducing and motivating the problem of attribution attacks – it is not the methodology or the results or the techniques (though they are very interesting in their own right) – this has opened the doors to future works such as attacking other attribution methods and more complex models. It wasn’t clear how the problem should be formulated or if there is a computationally feasible way of attacking influence functions or if the attacks will succeed even in simpler settings.
>
> Additionally, we have extra contributions such as an impossibility theorem (Thm. 1) which points to the non-triviality of the problem and an efficient way of backpropagating through influence functions or inverse-hessian-vector-products – which is of independent technical interest.
>
> d) **Example for a real world scenario where our threat model is feasible** : Note that the influence calculator is also compensating data providers. There could be two very natural scenarios for why it is the attacker 1) The influence calculator/model trainer is incentivized financially and wants to save money. Therefore it wants to artificially reduce the compensation it pays to the data providers, which can be done by reducing influence scores. Note that our proposed attacks are symmetric and can be used to both increase or decrease the data value. 2) Some data providers have colluded with the influence calculator to increase their own compensation (while reducing that of others); this is motivated by the fact that it may not be possible to easily change the data (such as in a biological domain, it may not be possible to change the DNA of a sample). This is mentioned in L176-178 and L183 in the paper.
>
> e) **L2 norm & dist function**: Note the direction of our condition : we are saying that if the L2 distance between two sets of parameters is small, then their performance is close. Essentially we want to look for modified weights within a C-ball of the original weights. For any reasonable function which does not change abruptly, this condition should hold and give us a set of weights with good accuracy. Your example is that of large distance change but the performance being the same.
>
> Having clarified the condition, there could be other ways of encoding the condition for different models and should be explored by the adversary based on their model architecture.

---

> > ### Author Response · Authors · 2024-11-20
> >
> > f) **Parameter C**: Note that C is a hyperparameter set by the adversary itself. C creates a tradeoff between the attack success rate and the accuracy loss. Greater the accuracy loss higher are the chances of the attack not being discreet and the adversary getting caught.  Hence C is chosen by the adversary based on how risk-friendly (strongly incentivized) or risk-averse (weakly incentivized) it is.
> >
> > To find C one can start with a small value of C and do a simple line (or any other) search from there. A powerful adversary will be able to do a more fine-grained search for C.
> >
> > g) **Eq. 3** : We are interested in an estimate of the net influence of the training sample over multiple randomly sampled test points rather than a single test point, to get rid of the variance. Yes, canceling out occurs.
> >
> > We hope our answers addressed your concerns. Please let us know if there is anything more we could clarify or address. We look forward to hearing from you!
> >
> > Cheers,
> >
> > Authors

---

> > > ### Author Response · Authors · 2024-11-22
> > >
> > > Dear Reviewer,
> > >
> > > Hope you are doing well!
> > >
> > > As we are getting closer to the weekend and thanksgiving week, we wonder if you got a chance to look at our response. Looking forward to hearing from you!
> > >
> > > Regards,
> > >
> > > Authors

---

### Official Review · Reviewer_NLFU · 2024-11-02

**Soundness:** 3
**Presentation:** 2
**Contribution:** 2
**Rating:** 5
**Confidence:** 5

**Summary:**

This paper investigates the vulnerability of influence functions in adversarial settings. Influence functions are commonly used to assess the contribution of training data to model predictions, but they may be susceptible to manipulation in incentivized applications like data valuation and fairness. The authors propose two types of attacks: a targeted attack designed to increase the influence score of specific samples to raise their data valuation, and an untargeted attack that alters influence scores through simple model weight scaling, affecting model fairness. Experimental results show that these attacks can be effectively implemented on multiclass logistic regression models and standard datasets without significantly compromising model accuracy. Additionally, the paper presents an optimization method for inverse computation of influence functions to enhance memory and time efficiency. Finally, the authors propose possible defense strategies and highlight the limitations of influence functions in adversarial environments.

**Strengths:**

1. I find the exploration of weaknesses in influence attribution an interesting phenomenon with practical application support, as highlighted in the proposed applications.
2. The authors provide theoretical analysis, demonstrating that under certain conditions, the influence scores of specific data samples cannot be manipulated.

**Weaknesses:**

1. The paper references the instance attribution method from [1] (which relies on local model parameters), but there are currently many new instance attribution methods (incorporating model parameters from different training stages). The paper should at least discuss these instance attribution methods (as in "Training data influence analysis and estimation: a survey") to rule out that the vulnerabilities are due to flaws in the method itself.
2. Many logical details are placed in the appendix, which makes reading somewhat challenging. I believe it would be more reasonable to include core logic in the main text and place some logical explanations in the appendix, especially Section A.1 (this does not affect my scoring decision).
3. From reading the application, I can clearly understand the significance of this task, but I cannot precisely grasp how to achieve such a goal. The last formula on page 5 appears to be an intuitive attack method without proof of achieving the intended objective.
4. I believe the research task itself is meaningful but lacks discussion on the more advanced instance attribution methods [2]. If these methods could be included in the discussion, I would be more inclined to raise my score.
[1] Pang Wei Koh and Percy Liang. Understanding black-box predictions via influence functions. In International Conference on Machine Learning, pp. 1885–1894. PMLR, 2017
[2] Training data influence analysis and estimation: a survey

**Questions:**

Please refer to weaknesses.

---

> ### Author Response · Authors · 2024-11-20
>
> Dear Reviewer,
>
> Thanks for taking the time to review our paper, we deeply appreciate it.
>
> We are very glad that you find the idea of exploring vulnerabilities in influence functions and our theoretical analysis interesting. Next we address your concerns.
>
> a) **Add a discussion**: We agree with you that the vulnerabilities are due to flaws in the method itself and *would be very happy to add a discussion with the papers you mentioned.*
>
> b) **Formula on Page 5**: As mentioned in L263-269, this formula is meant to capture a baseline for increasing the influence of a target sample without using influence functions. A natural way to do this is to put more importance on the target sample in the loss function, which means the updates will be guided more by the loss on the target sample (eq.5). This can be achieved by either directly reweighing the loss function with different alphas or resampling the target point multiple times in the batch. We found the former to be unstable during training and stuck to the later.
>
> This is the simplest and most intuitive non-influence baseline we could think of. Please let us know if things are still unclear and we will try our best to clarify.
>
> We hope our answers addressed your concerns. Please let us know if there is anything more we could clarify or address. We look forward to hearing from you!
>
> Cheers,
>
> Authors

---

> > ### Author Response · Authors · 2024-11-22
> >
> > Dear Reviewer,
> >
> > Hope you are doing well!
> >
> > As we are getting closer to the weekend and thanksgiving week, we wonder if you got a chance to look at our response. Looking forward to hearing from you!
> >
> > Regards,
> >
> > Authors

---

### Official Review · Reviewer_faoh · 2024-11-04

**Soundness:** 1
**Presentation:** 2
**Contribution:** 1
**Rating:** 1
**Confidence:** 4

**Summary:**

The paper proposes an adversarial attack where an adversary seeks to train a malicious logistic regression model which attains similar performance to the original model but has a different distribution of influence estimates. The authors conduct experiments for the downstream tasks of data valuation and fairness.

**Strengths:**

- The paper conducts experiments on multiple datasets.
- The concept of adversarial attacks on influence scores is an interesting idea overall.

**Weaknesses:**

In my opinion, the paper possesses major limitations that invalidate its contributions. My major concern revolves around the proposed threat model itself, which seems untenable for real-world practical applications. Other issues are concerned with limited model and dataset evaluation, and the overall simplicity of the work:

- **Unjustifiable Threat Model**: The threat model makes multiple assumptions about the influence (and data valuation) problem pipelines in order to make the attack viable. First, it assumes that the data is a trivial commodity and that this will be provided as is by the data providers to an influence "calculator" who themselves are the adversary. I find this to be highly implausible-- data is generally protected and it is more likely for the data owners / model trainers to be benign as model training is a computationally costly process. It seems highly unlikely (especially if the influence calculator can potentially be an adversary) that the data providers will provide highly sensitive data to this third-party as its outputs cannot be guaranteed to be trustworthy (i.e. anyone can train a model as they would like-- why would they trust them and their outputted scores?). Second, the threat model assumes that the influence calculator who is training the model possesses malicious intent but the reasons for this are unclear. As the influence calculator is taking on the overhead of model training, what is their incentive in doing so? They are training the model to achieve similar performance on the test set (as the paper also measures) and hence, are likely going to serve it for potential production use. Why would they change the influence scores for their own model to compensate data providers differently while incurring the cost of another model retraining then? Their main incentive should lie in ensuring robust model performance for end-users. Perhaps these aforementioned issues can be justified with some real-world examples or use-cases, but it is not motivated sufficiently in the current version of the paper and I do not think this setting is tenable.
- **Limited Models and Datasets**: The paper only considers a simple logistic regression model in experiments and all the datasets are tabular datasets obtained via embeddings from deeper models. Why is logistic regression a good base model for data valuation when the task itself might require a deep learning model that is non-convex? Overall, this is very restrictive as a setting and the paper needs to consider end-to-end models for experiments.
- **Simplicity of Attack**: Owing to the assumptions made in the threat model, the attack is very simple-- the approach just requires training a model with an adversarial objective while putting a constraint on performance. This is technically quite trivial and somewhat lower than the bar for ICLR.

**Questions:**

Please see the weaknesses listed above-- each can be considered as a question.

---

> ### Author Response · Authors · 2024-11-20
>
> Dear Reviewer,
>
> Thanks for taking the time to review our paper, we deeply appreciate it.
>
> We are glad that you find the idea of attacking influence functions interesting. Next we address your concerns.
>
> a) **Threat model**: To begin with, the point is that – there is no one universal threat model or attack which is applicable in all settings. Different circumstances call for different threat models and attacks.
>
> *“It seems highly unlikely (especially if the influence calculator can potentially be an adversary) data providers will provide highly sensitive data to this third-party”*: There are multiple assumptions in this statement. Firstly, it is assumed that all data is highly sensitive and that providers won’t be sharing this data. In contrast, the idea of data valuation is set in an AI economy where different users are “ready” to sell their data for appropriate compensation and our paper targets such settings. We provide a way this compensation can be manipulated by an adversary and data providers will get a different compensation from what their data is worthy of. We suggest you to read [1,2] to get a better sense of data valuation if interested. Secondly, before our paper, it wasn’t even known that there could be an adversarial setting for influence-based attributions or data valuation at all! This is the point of our paper – to bring to light the potential adversarial actors in this setting. Our paper is an important tale of caution for these very reasons!
>
>
> *Real world scenarios for influence calculator as the attacker*: Note that the influence calculator is also compensating data providers. There could be two very natural scenarios for why it is the attacker 1) The influence calculator/model trainer is incentivized financially and wants to save money. Therefore it wants to artificially reduce the compensation it pays to the data providers, which can be done by reducing influence scores. Note that our proposed attacks are symmetric and can be used to both increase or decrease the data value. 2) Some data providers may have colluded with the influence calculator to increase their own compensation (while reducing that of others); this is motivated by the fact that it may not be possible to easily change the data (such as in a biological domain, it may not be possible to change the DNA of a sample). This is mentioned in L176-178 and L183 in the paper.
>
> *“another model retraining then”*: There is perhaps a slight misunderstanding. The attacked model is not retrained from scratch – few updates are made on top of the original honest model.
>
> b) **Linear Models & Simplicity** : Note that ours is the first paper which introduces the problem of attacking attributions and proposes attacks with in-depth analysis. We believe the most important contribution of the paper is introducing and motivating the problem of attribution attacks – it is not the methodology or the results or the techniques (though they are very interesting in their own right) – this has opened the doors to future works such as attacking other attribution methods and more complex models. It wasn’t clear how the problem should be formulated or if there is a computationally feasible way of attacking influence functions or if the attacks will succeed even in simpler settings.
>
> The attacks you call simple do not have a 100% attack success rate on even linear models across multiple datasets, which points to the non-triviality of the problem and leads to a very interesting impossibility result (Thm. 1). It is a very interesting theorem in our opinion!
>
> Additionally, we have extra contributions such as an efficient way of backpropagating through inverse-hessian-vector-products – which is of independent technical interest.
>
> We hope our answers addressed your concerns. Please let us know if there is anything more we could clarify or address. We look forward to hearing from you!
>
> Cheers,
>
> Authors
>
> [1]Inflow, outflow, and reciprocity in machine learning.  Sundararajan et. al. ICML 2023
>
> [2]Towards efficient data valuation based on the shapley value. Jia et.al. AISTATS 2019 (Influence functions are used as the value estimating functions.)

---

> > ### Comment · Reviewer_faoh · 2024-11-20
> >
> > Thank you for your response. I am very familiar with the literature on data valuation. The issue is your threat model is not justifiable in any practical scenario, there are way too many assumptions being made that do not hold in the real-world to "create" this attack scenario. Reviewers PY2G, jono, and 71Su also share the exact same concerns. For a useful paper in this space, the attack should have some real-world significance and as that doesn't hold here, I am not convinced. Thank you for your efforts but unfortunately, my concerns are not addressed and I will maintain my current score.

---

> ### Author Response · Authors · 2024-11-20
>
> Thanks for your response. Could you please point to exactly why the real-world scenarios given by us in the response don't convince you? We believe them to be pretty general and realistic.

---

> > ### Comment · Reviewer_faoh · 2024-11-20
> >
> > Dear authors, I cannot come up with one realistic (i.e. real-world) example where the influence calculator is open to malicious (re)training and the data providers are completely benign and the threat model can be justified. Given the obvious option for model manipulation (even without influence scores being manipulated) this does not seem like an incentive structure that anyone will opt in to (i.e. why would benign data providers do this knowing the model can be maliciously adjusted in any way?). While there are a number of hypothetical arguments provided in the rebuttal, I have not seen a real-world example that provides evidence to the contrary. Thanks.

---

> ### Author Response · Authors · 2024-11-20
>
> Dear Reviewer, thanks for your engagement.
>
> The reviewer’s argument is an oversimplification in our opinion, for the following reason.
>
> While it is known that different models can have similar accuracy, it was not known before our paper that a model can be *steered* to have vastly different and *desired* influence rankings while maintaining similar accuracy. As an attack paper *this is* the vulnerability we are exposing. Do you any literature in mind which shows that the model can be maliciously adjusted to produce *desired* influence rankings?
>
> *After this paper* it is clear that the benign data providers (though not all have to be benign from our example 2) need some guarantees on the training process. How would this point be made in the absence of our paper?
>
> Cheers, Authors

---

> > ### Comment · Reviewer_faoh · 2024-11-20
> >
> > Happy to discuss. The point you are missing is, if I control the model (i.e. influence calculator), then why would a benign data provider buy in to this mechanism? This is beyond the influence manipulation claim, if you cannot trust the model provider and there is no way of checking for all possible manipulations, the model provider can manipulate the model as they would like (for e.g. if they want, they will figure out a way to change influence scores, sure). But why will any benign individual opt in to this setting? This threat model cannot be justified and the claim you are making (safeguards needed for data providers) is made in the absence of this work (because models can always be manipulated; e.g., backdoor attacks) . Can you provide a real-world example that justifies this attack? That is all I am asking for.
> >
> > Second, your question to me _do you have any literature in mind which shows that the model can be maliciously adjusted to produce desired influence rankings?_ is posed in a way that obfuscates our point of discussion. Of course there will not be any papers under this threat model (that you propose) that will show manipulation of influence scores, but the absence of this work does not mean that there are no issues with the threat model and attack setting. I have yet to see some evidence for why this threat model can occur in the real-world. Thanks.

---

> ### Author Response · Authors · 2024-11-20
>
> Dear Reviewer, thanks for your engagement.
>
> The bone of contention here is that you *assume* if the adversary controls the model training process they will be able to produce desired influence scores (“because models can always be manipulated”,”if they want, they will figure out a way to change influence scores, sure”). You assume that attacking influence functions is a trivial problem — this is where we disagree — our paper is about creating this attack. Why is it obvious that one can produce two models with similar accuracies but desired influence rankings? (this is the literature we were asking for). This is not obvious for even feature-based attributions [1]. The fact that we don’t get 100% attack success rate in our experiments and our impossibility theorem point to the non-triviality of this problem. The claims we make for safeguarding for data providers *cannot be made independently of our work* just because there exist many attacks in the literature — none of which attack influence functions in particular and whether they can or not is an open question without this work.
>
> We don’t think there is discussion or emphasis in data valuation papers that a trusted model provider is needed. With our paper we make a concrete case that for benign data providers either a trusted model provider or a proof of honest training is required  (this is mentioned in our discussion section).
>
> Now answering your question of *why hiding training process from the data providers is realistic* —  We believe this is the usual setting in the real world. Consider a company such as OpenAI or a bank. They buy data from different vendors to build their models*. But do they give the model or show the training process to the data providers? No, the cord is cut at buying the data and the model and its training is proprietary.
>
> Cheers,
>
> Authors
>
> [1] Fairwashing Explanations with Off-Manifold Detergent ICML 2020
>
> *Currently, these deals might be done with negotiation but is subject to change in the future as the field of data valuation progresses.

---

> > ### Comment · Reviewer_faoh · 2024-11-20
> >
> > Thank you for your response. I believe we are not making headway because we are not talking about the same things. Starting from first principles, an attack only makes sense if the threat model is tenable and justifiable. Irrespective of the contributions (i.e. the results of the paper showing that influence scores can be manipulated) if this foundational aspect does not hold, the attack by itself is not useful. This is my point.
> >
> > Regarding the claim that hiding models is the status quo, why would OpenAI itself try to pay certain vendors more than others? There is no incentive mechanism as such. If they want data from certain vendors over others then they should just take data from those vendors. From the perspective of the data provider: in the future you are referencing, for them to assess their data's worth  there need to be appropriate incentive mechanisms in place that incentivize them to contribute data. The data's value/influence needs to be independently verifiable. Otherwise they should just be happy with whatever money OpenAI pays them and there is no point to data valuation.

---

> ### Author Response · Authors · 2024-11-22
>
> Dear Reviewer,
>
> Thanks for your engagement.
>
> We believe our threat model is tenable, due to the reasons mentioned in the previous comments. We respect your point of view but would like to end the discussion as of now as we do not see a common ground being achieved.
>
> Cheers,
>
> Authors

---

### Official Review · Reviewer_71Su · 2024-11-04

**Soundness:** 2
**Presentation:** 3
**Contribution:** 2
**Rating:** 3
**Confidence:** 3

**Summary:**

This paper examines the potential for manipulating influence-based attributions. The authors explore two use cases: data valuation and fairness. In data valuation, an attacker aims to inflate the influence scores of specific data samples to increase their perceived value. In fairness, the attacker seeks to reduce the fairness of a downstream model by manipulating influence scores used for data reweighing. The paper presents targeted and untargeted attack algorithms, demonstrating their effectiveness on logistic regression models trained on ResNet features and standard fairness datasets. The authors also present a theoretical impossibility theorem.

**Strengths:**

- The paper is well-written and well-organized.
- The paper provides a practical and efficient algorithm to compute the backward pass through Hessian-Inverse-Vector Products for influence-based objectives.
- The experimental results in the sources show that influence-based attacks, both targeted and untargeted, can successfully manipulate influence scores while maintaining model accuracy.

**Weaknesses:**

- The setting of this working is confused to me. It presents a scenario where an adversary can manipulate the model training process to achieve desired influence scores. Is it a realistic scenario in practice and why would an adversary manipulate the influence score when he could manipulate the model itself?
- The method aims to manipulate the influence scores while maintaining the similar test accuracy, and this accuracy is measured by a `dist` function between the parameters and limited by a radius value $C$. However, the choice of $L_2$ norm to measure the distance between two high-dimensional vectors is not always consistent. For example, one can construct another network by multiplying all the parameters by $10^6$ and adding a normalization on each layer, which gives a same result but the $L_2$ distance will be much different. The authors should provide more justification for this choice.
- The experiments described in the paper primarily focus on logistic regression, which may not be representative of more complex models. Will the proposed attack methods be effective on more complex models? Is the objective function defined based on the `dist` function still valid due to the above reason?

**Questions:**

See weaknesses

---

> ### Author Response · Authors · 2024-11-20
>
> Dear Reviewer,
>
> Thanks for taking the time to review our paper, we deeply appreciate it.
>
> We are glad that you find the paper well-written and well-organized, that you appreciate our practical and efficient backpropagation algorithm and our experimental results. Next we address your concerns.
>
> a) **Setting of our attack**: There seem to be two major concerns you have in this regard. Firstly, why would the influence calculator manipulate the model when it can train it itself. The point is that the model arrived at with naive training may not have the desired influence scores. Therefore the model needs to be steered such that it gives the desired influence rankings, which is what our attack does.
>
> Secondly, what is a realistic scenario where the model training/influence calculator attacks? There could be two scenarios 1) The influence calculator/model trainer is incentivized financially and wants to save money. Therefore it wants to artificially reduce the compensation it pays to the data providers, which can be done by reducing influence scores. Note that our proposed attacks are symmetric and can be used to both increase or decrease the data valuation. 2) Some data providers have colluded with the influence calculator to increase their own compensation (while reducing that of others); this is motivated by the fact that it may not be possible to easily change the data (such as in a biological domain, it may not be possible to change the DNA of a sample).
>
> b) **Dist function**: The attacker is free to choose whatever method it deems fit for manipulating the model to get desired influence rankings while maintaining accuracy. Along these lines, the attacker can choose any function for implementing dist. We chose the L2 norm for experiments because of its simplicity and ease when using it in Alg. 3 and this choice serves us well in terms of attack success rates!
>
> Note that the attacker wants to maintain accuracy “while getting the desired influence rankings.” It is not clear the method you propose could attain desired influence rankings and is a question for experimentation. Also note that no matter what choice of dist we made, one could always why? So one can also ask why the suggested function is better than the L2 norm? – answers to these questions depend on the particular use-case.
>
> As a side note, in your suggested function, you make an additional implicit assumption that it is allowed to change the model architecture between the original and attacked models (which may not be the case when the influence calculator is commissioned by a client to train a model for a particular architecture or when there are standard architectures leading to optimal performance – so a deviation from the standardarchitecture can be caught by an auditor or when fine-tuning on existing model architectures) and multiplying all parameters by a constant, especially large constants, can be easily caught by an auditor.
>
> c) **Complex Models** : More complex models are certainly interesting. Based on our intuition of working with influence functions, we do believe that the chances of the attacks working are good since these functions are sensitive to model weights. Regardless, please note that ours is the first paper which introduces the problem of attacking attributions and proposes attacks with in-depth experiments. We believe the most important contribution of the paper is introducing and motivating the problem of attribution attacks – this has opened the doors to future works such as attacking more complex models as you mentioned. It wasn’t clear how the problem should be formulated or if there is a computationally feasible way of attacking influence functions or if the attacks will succeed even in simpler settings. Given this fact, we proceeded with linear models. Even for linear models and multiple datasets, we do not get a 100% attack success rate, which points towards the non-triviality of the problem and leads to a very interesting impossibility result (Thm. 1).
>
> Additionally, we have extra contributions such as an efficient way of backpropagating through inverse-hessian-vector-products – which is of independent technical interest.
>
> We don’t see why the L2 norm would not be a valid choice for dist function in this setting. Regardless, it is up to experimental analysis.
>
> We hope our answers addressed your concerns. Please let us know if there is anything more we could clarify or address. We look forward to hearing from you!
>
> Cheers,
>
> Authors

---

> > ### Author Response · Authors · 2024-11-22
> >
> > Dear Reviewer,
> >
> > Hope you are doing well!
> >
> > As we are getting closer to the weekend and thanksgiving week, we wonder if you got a chance to look at our response. Looking forward to hearing from you!
> >
> > Regards,
> >
> > Authors

---

### Official Review · Reviewer_PY2G · 2024-11-04

**Soundness:** 2
**Presentation:** 2
**Contribution:** 1
**Rating:** 3
**Confidence:** 4

**Summary:**

The paper studied a setup in which the influence scores of training samples can be manipulated through a modified training procedure by changing the loss function to maximize the influence scores of target samples on a subset of test samples. The paper aims to show that there could be vulnerabilities in influence scores that affect downstream applications, such as those related to fairness.

**Strengths:**

1. The paper is easy to follow, and the results and methodology are clearly presented in tables and figures/diagrams.

2. The overall idea of studying vulnerabilities in influence functions is interesting, especially as there are some applications built on using influence scores. However, I have some concerns, which are discussed in the following points.

**Weaknesses:**

1. The first and most important point to raise is that the attack setup does not seem practical in the real world. The paper provides a framework that includes two components: 1) Data Provider and 2) Influence Calculator. It is assumed that the data provider is truthful, while the Influence Calculator, which trains a "model" and calculates the influence scores, is maliciously manipulated. I can't think of a real-world scenario in which an Influence Calculator would deliberately aim to manipulate influence scores or would need to alter the influence score to indirectly affect the desired behavior, especially when it can directly train a model and is considered the target (or deployed) model. The reverse scenario seems more plausible, where a Data Provider among all providers aims to change the influence score distribution based on its own interests. Although the paper attempts to justify a scenario involving "fairness manipulation," this does not seem plausible to me. To modify the fairness of a model, it would need to be trained in a way that specifically adjusts the influence scores; however, there are more straightforward ways to alter this dynamic, specially when the training procedure can be altered.

2. Another key issue is that altering the influence scores distribution requires access to the test set to manipulate the top-k scores. This approach is problematic, as it risks causing the modified influence scores to overfit to the specific test set. This tendency is evident in Figure 2, where the "transfer" results (that uses held out test set) are notably less impactful, suggesting that the changes in influence scores are highly dependent on the specific test set used in the optimization process.

3. While the paper discusses it, there seems to be a key limitation in the "Multi-Target Attack" setup: modifying the influence scores for a subset of samples is not an independent process for each sample. The authors attempt to mitigate this by employing multiple random initializations; however, this solution appears to be costly.

4. The current version of the method appears to be computationally feasible only for linear models, which further limits the practicality of the methodology (in addition to my first point). However, there has been significant progress in making influence score calculations feasible for non-linear and large models, such as in "TRAK: Attributing Model Behavior at Scale" (https://arxiv.org/abs/2303.14186), which are more compelling for study.

5. The writing style is not ideal; too many paragraphs begin with a bold sentence or title. While this may be acceptable to some readers, I found it somewhat confusing to determine what is truly important. I would suggest that the authors use bold text sparingly, reserving it for points that are especially significant.

**Questions:**

1. Regarding the first point mentioned above, how practical is this scenario in the real world?

2. Can changes in the influence scores be achieved on the data provider's side without needing access to the model or altering the training procedure? I assume this might be related to adversarial attacks and data poisoning literature; however, it would be beneficial to discuss this from the perspective of influence functions.

3. How does the scale of the data affect the difficulty of the task? Based on Figure 2, it appears that as the training data size increases, the task of changing the influence scores becomes more challenging. Given the trend that more data generally improves performance, does a larger dataset also make influence score modification more difficult?

---

> ### Author Response · Authors · 2024-11-20
>
> Dear Reviewer,
>
> Thanks for taking the time to review our paper, we deeply appreciate it.
>
> We are glad that you find the paper easy to follow, clear and also find the idea of studying vulnerabilities in influence functions interesting. Next we address your concerns.
>
> a) **Attack setup not real?** : There seem to be three major concerns you have in this regard. Firstly, why would the influence calculator manipulate the model when it can train it itself. The point is that the model arrived at with naive training may not have the desired influence scores. Therefore the model needs to be steered such that it gives the desired influence rankings, which is what our attack does.
>
> Secondly, what is a realistic scenario for the model training/influence calculator attack? There could be two scenarios 1) The influence calculator/model trainer is incentivized financially and wants to save money. Therefore it wants to artificially reduce the compensation it pays to the data providers, which can be done by reducing influence scores. Note that our proposed attacks are symmetric and can be used to both increase or decrease the data valuation. 2) Some data providers may have colluded with the influence calculator to increase their own compensation (while reducing that of others); this is motivated by the fact that it may not be possible to easily change the data (such as in a biological domain, it may not be possible to change the DNA of a sample). For attacks by the data provider, please read the next point.
>
> Thirdly, about the plausibility of fairness manipulation. This attack is feasible in scenarios when the methodology we attack is already in use – of using influence functions to weigh samples and train a downstream model as proposed by [1]. Certainly there can exist other ways to alter the fairness of a model, but ours is an influence function paper and we wish to show how different usages of influence functions can be attacked. A simple takeaway from the success of our fairness attack is that perhaps using influence functions to lead to downstream fair models is not right-minded due to the existence of our attack.
>
> Additionally, just want to clarify a possible misunderstanding – we do not have control over training the downstream fair model, we can only train the base model used to find the weights (influence scores) of training samples. Please see Fig.8 in the Appendix for a pictorial representation of the flow.
>
> b) **Attacks from the Data provider** : To begin with, the point is that – there is no one universal threat model or attack which is applicable in all settings. Different circumstances call for different attacks and threat models.
>
> Now whether attacks can be done at the data providers’ end or not is an interesting question and needs further investigation. Intuitively, data poisoning could be a plausible technique but the most important question is – since the data poisoning should take place prior to model training, what information can  the data provider use in the absence of the trained model? Second important question is – since we are talking of relative rankings, how can a data provider raise its own data value wrt others or drop the value for others without access to the data of other providers? Some careful thought needs to go into these questions. Whether or not data poisoning attacks will work for the case of changing influence scores (rather than labels) and at the same time be computationally feasible, can only be confirmed with experimentation and will lead to a different paper altogether. However, *we will be happy to add this discussion on data poisoning in the paper.*
>
> Having said this, our setting is relevant in cases where data poisoning attacks may not be appropriate – 1) where it is not possible to manipulate even your own data as we mentioned in the paper or 2) not possible to manipulate the data from others due to lack of access. For 1), consider datasets made from DNAs – it is not possible to arbitrarily change data in such settings and also depends on how the DNA (or the data) is collected in terms of source of origin. For 2), a data provider may conduct data poisoning attacks on its own data, but how will it change the data for others to rise relatively in rankings when it doesn’t have access to their data?
>
> c) **Overfitting to the test set?** : This question is more nuanced than it seems, but there can be simple approaches to prevent overfitting such as stopping early. Now coming to the nuance, since influence scores have to be calculated on test samples, one would always need a test set. Whether your concern is a problem or not depends on where the test set is coming from.

---

> ### Author Response · Authors · 2024-11-20
>
> If an external checker/auditor is verifying the influence scores, the influence calculator will show its test set and the calculated influence scores to the checker/auditor since influence scores are tied to the test sets and there is no established consensus or study from our knowledge which measures how much influence scores are sensitive to the test set. If an external party which will use influence scores is supplying the test set to the influence calculator, the influence calculator may overfit. If an external party which will use influence scores keeps a separate test set to check the influence scores, then the influence calculator should use techniques to stop early, while still providing its own test set to the party to show that its calculated influence scores are consistent with the test set it used and prevent getting caught.
>
> d) **"Multi-Target Attack" setup**: We agree with you, but on the flip side, our greedy attack is the simplest thing one could do (without considering complex training interactions) and yet we get good success rates! This means that our attack is still powerful! Though there is room to see if complicated techniques can do any better.
>
> e) **Scale of the data**: Increasing the data for which influence scores are to be calculated does make the task harder if we are concerned with top-k/bottom-k rankings. However, our attack can still apply when (1) considering fine-tuning with smaller datasets for specific domains (where the task of data valuation becomes more relevant) or (2) when we consider societal application tasks in finance or healthcare which have smaller datasets, or (3) when we care about altering the absolute value of influence for some data samples.
>
> f) **More complex models and other attribution methods**: Please note that ours is the first paper which introduces the problem of attacking attributions and proposes attacks with in-depth experiments. We believe the most important contribution of the paper is introducing and motivating the problem of attribution attacks – this has opened the doors to future works such as attacking other attribution methods as you mentioned. It wasn’t clear how the problem should be formulated or if there is a computationally feasible way of attacking influence functions or if the attacks will succeed even in simpler settings. Given this fact, we proceeded with linear models. Even for linear models and across multiple datasets, we do not get a 100% attack success rate, which points towards the non-triviality of the problem and leads to a very interesting impossibility result (Thm. 1).
>
> Additionally, we have extra contributions such as an efficient way of backpropagating through inverse-hessian-vector-products – which is of independent technical interest.
>
> Certainly applicability to more complex models and manipulating other methods (though we believe studying influence functions is also very important given the huge number of use-cases and applications proposed for them) *are both interesting questions and have been mentioned by us in the conclusion as future works.*
>
> We hope our answers addressed your concerns. Please let us know if there is anything more we could clarify or address. We look forward to hearing from you!
>
> Cheers,
>
> Authors
>
> [1] Achieving fairness at no utility cost via data reweighing with influence. Li et.al. ICML 2022

---

> > ### Author Response · Authors · 2024-11-22
> >
> > Dear Reviewer,
> >
> > Hope you are doing well!
> >
> > As we are getting closer to the weekend and thanksgiving week, we wonder if you got a chance to look at our response. Looking forward to hearing from you!
> >
> > Regards,
> >
> > Authors

---

> > > ### Comment · Reviewer_PY2G · 2024-11-27
> > >
> > > Dear Authors,
> > >
> > > Thank you for your effort in addressing the concerns. However, after reviewing your response, I still believe that the main issue with this attack lies in its failure to consider a plausible scenario, as also pointed out by other reviewers. Consequently, I will need to maintain my current score.

---

> > > > ### Author Response · Authors · 2024-11-27
> > > >
> > > > Dear Reviewer,
> > > >
> > > > Could you please explain why the scenarios we gave in the response above are not plausible in your opinion?

---

### Meta-Review · Area_Chair_Y8ED · 2024-12-05

**Metareview:**

I have read all the materials of this paper including the manuscript, appendix, comments, and response. Based on collected information from all reviewers and my personal judgment, I can make the recommendation on this paper, reject. No objection from reviewers who participated in the internal discussion was raised against the reject recommendation.

**Research Question**

The authors study the manipulation of data valuation. The data valuation depends on the learning algorithm, where the authors assume the data and learning algorithm are controlled by different agents. The targeted scenario is unrealistic. Thus, there is no necessary to comment other parts of this paper.

The review team including all reviewers and me, believes this paper does not meet the bar of ICLR.

**Additional Comments On Reviewer Discussion:**

All reviewers participated into the internal discussion and supported the rejection recommendation.

The authors attempted to address this in rebuttal, all reviewers and I did not find this convincing. The data valuation should be associated with the learning algorithm, i.e., the same sample might have different impacts based on difference algorithms. Therefore, separation of data and the learning algorithm makes the setting unrealistic.

---

### Decision · Program_Chairs · 2025-01-22

Reject